# Sparse Attention Decomposition Applied to Circuit Tracing

## Abstract

Many papers have shown that attention heads work in conjunction with each other to perform complex tasks. It's frequently assumed that communication between attention heads is via the addition of specific features to token residuals. In this work we seek to isolate and identify the features used to effect communication and coordination among attention heads in GPT-2 small. Our key leverage on the problem is to show that these features are very often sparsely coded in the singular vectors of attention head matrices. We characterize the dimensionality and occurrence of these signals across the attention heads in GPT-2 small when used for the Indirect Object Identification (IOI) task. The sparse encoding of signals, as provided by attention head singular vectors, allows for efficient separation of signals from the residual background and straightforward identification of communication paths between attention heads. We explore the effectiveness of this approach by tracing portions of the circuits used in the IOI task. Our traces reveal considerable detail not present in previous studies, shedding light on the nature of redundant paths present in GPT-2. And our traces go beyond previous work by identifying features used to communicate between attention heads when performing IOI.

## 1 Introduction

Recent work has made progress interpreting emergent algorithms used by language models in terms of circuits (Olah et al., 2020). Much of the work in model interpretability views a model as a computational graph (Geiger et al., 2021), and a circuit as a subgraph having a distinct function (Conmy et al., 2023; Wang et al., 2023; Marks et al., 2024). In language models, the computation graph is typically realized through communication between model components via the residual stream (Elhage et al., 2021).

In this context, an important subtask is *tracing a circuit* (Wang et al., 2023; Lieberum et al., 2023; Conmy et al., 2023) – identifying causal communication paths between model components that are significant for model function. Focusing on just communication between attention heads, one approach to circuit tracing might be to track information flow. For example, given an attention head computing a score for a particular pair of tokens, one might ask which upstream heads modify those tokens in a way that functionally changes the downstream attention head's output. A straightforward attack on this question would involve direct inspection of residuals and model component inputs and outputs at various places within the model. Unfortunately and not surprisingly, direct inspection reveals that the great majority of upstream attention heads meet this criterion. Each downstream attention head examines a pair of query and key subspaces, and most upstream heads write at least some component into each downstream head's query or key subspace. The problem this causes is that, absent interventions using counterfactual inputs, it is not clear which of those contributions are making changes that are significant in terms of model function.

Many previous studies have approached this problem by averaging over a large set of inputs, comparing test cases with counterfactual examples, and by using interventions such as patching (Zhang & Nanda, 2024; Goldowsky-Dill et al., 2023). This approach has many successes, but it can be hard to isolate specific communicating pairs of attention heads and hard to identify exactly what components of the residual are mediating the communication.

In this paper we explore a different strategy. We ask whether leverage can be gained on this problem by looking for low dimensional components of each upstream head's contributions. To do so, we ask: is there a change of basis associated with any given attention head, such that components of its input in the new basis make sparse contributions to the attention head's output? If so, those components may represent the most significant parts of the input in terms of model function. Our main result is to find a change of basis for the attention head's inputs, such that attention scores are sparsely constructed in the new basis. We show that this basis comes from the singular value decomposition of the QK matrix of the attention head.

Our testbed is GPT-2 small applied to the Indirection Object Identification (IOI) task (Wang et al., 2023). In this setting we show that in the new basis, tracing inputs to attention heads back to sources serves generally identifies a small set of upstream attention heads whose outputs are sufficient to explain the downstream attention head's output. Further, the small set identified generally includes upstream heads that are known to be functionally associated with the downstream head. We also show that, in the new basis, the representations of tokens can be correlated with semantic features. We exploit these properties to build a communication graph of the attention heads of GPT-2 for the IOI task. By construction, each edge represents a causal direct effect between an upstream head and the attention score output from the downstream head; more interestingly, we demonstrate that the edges in this graph generally identify a communication path that has causal effect on the ability of GPT-2 to perform the IOI task.

In sum, our contributions are twofold. First, we draw attention to the fact that attention scores are typically *sparsely decomposable* given the right basis. This has significant implications for interpretability of model activations. Note that this is very different from the uses of SVD to analyze model matrices in previous work; we are *nowt* concerned with static analysis of model matrices themselves. Second, we demonstrate that by leveraging the sparse decomposition to denoise inputs to heads, we can effectively and efficiently trace functionally significant causal communication paths between attention heads.

## 2 RELATED WORK

The dominant style of circuit tracing is via *patching* (Zhang & Nanda, 2024; Goldowsky-Dill et al., 2023). That strategy has shown considerable success (Wang et al., 2023; Conmy et al., 2023; Hanna et al., 2023; Lieberum et al., 2023) but is time-consuming, generally requires the creation of a counterfactual dataset to provide task-neutral activation patches, may miss alternative pathways (Makelov et al., 2023; Mueller, 2024), and has been shown to produce indirect downstream effects that can even result in compensatory self-repair (McGrath et al., 2023; Rushing & Nanda, 2024).

In this work, we trace circuits using only a single forward pass over the data, eliminating the need for counterfactuals and avoiding the problems of self-repair after patching. The authors in (Ferrando & Voita, 2024) trace circuits in a single forward pass, and argue that the approach is much faster than patching, and avoids dependence on counterfactual examples and the risk of self-repair. We derive the same benefits, but unlike that paper we leverage spectral decomposition of attention head matrices to identify the signals flowing between heads.

Like distributed alignment search (Geiger et al., 2024) we adopt the view that placing a neural representation in an alternative basis can reveal interpretable dimensions (Smolensky, 1986). However, unlike that work, we do not require a gradient descent process to find the new basis, but rather extract it directly from attention head matrices. Likewise, using sparse autoencoders (SAEs) the authors in (Gurnee et al., 2023; Marks et al., 2024) construct interpretable dimensions from internal representations; our approach is complementary to the use of SAEs and the relationship between the representations we extract and those obtained from SAEs is a valuable direction for further study.

We demonstrate circuit tracing for the IOI task in GPT-2 small, which has become a 'model organism' for tracing studies (Wang et al., 2023; Conmy et al., 2023; Ferrando & Voita, 2024). Like previous studies, one portion of our validation consists of recovering known circuits; however we go beyond recovery of those known circuits in a number of ways, most importantly by identifying signals used for communication between heads.

The use of SVD in our study is quite different from its previous application to transformers. SVD of attention matrices has been used to reduce the time and space complexity of the attention mechanism (Wu et al., 2023; Wang et al., 2024) and improve reasoning performance (Sharma et al., 2024), often leveraging a low-rank property of attention matrices. Our work does not rely on attention matrices showing low-rank properties. We use SVD as a tool to decompose the *computation of attention*; the leverage we obtain comes from the resulting sparsity of the terms in the attention score computation. Likewise, previous work has shown interpretability of the singular vectors of OV matrices and MLP weights — though not of QK matrices (Millidge & Black). We show here evidence that it is the *representations of tokens in the bases provided by the singular vectors* of QK matrices that show intepretability.

## 3 BACKGROUND

In the model, token embeddings are $d$-dimensional, there are $h$ attention heads in each layer, and there are $t$ layers. We define $r = \frac{d}{h}$, which is the dimension of the spaces used for keys and queries in the attention mechanism. In GPT-2 small, $d = 768$, $h = 12$, $t = 12$, and $r = 64$.

The attention mechanism operates on a set of $n$ tokens in $d$-dimensional embeddings: $X \in \mathbb{R}^{n \times d}$. Each token $\mathbf{x} \in \mathbb{R}^d$ is passed through two affine transforms given by $\mathbf{x}^\top W_K + \mathbf{b}_K^\top$, $\mathbf{x}^\top W_Q + \mathbf{b}_Q^\top$, using weight matrices $W_K, W_Q \in \mathbb{R}^{d \times r}$

and offsets $\mathbf{b}_K, \mathbf{b}_Q \in \mathbb{R}^r$. Then the inner product is taken for all pairs of transformed tokens to yield *attention scores.* More precisely:

$$
\begin{aligned}
A' & = & (XW_Q + \mathbf{1b}_Q^\top)(XW_K + \mathbf{1}b_K^\top)^\top \\
& = & XW_QW_K^\top X^\top + XW_Q\mathbf{b}_K\mathbf{1}^\top + \mathbf{1b}_Q^\top W_K^\top X^\top + \mathbf{1b}_Q^\top \mathbf{b}_K\mathbf{1}^\top
\end{aligned}
\tag{1}
$$

We can capture (1) in a single bilinear form by making the following definitions:

$$
\Omega = \begin{bmatrix} W_QW_K^\top & W_Q\mathbf{b}_K \\ \mathbf{b}_Q^\top W_K^\top & \mathbf{b}_Q^T\mathbf{b}_K \end{bmatrix}, \quad \tilde{\mathbf{x}} = \begin{bmatrix} \mathbf{x} \\ 1 \end{bmatrix}.
\tag{2}
$$

Then we can rewrite the score computation (1) as

$$
A'_{ij} = \tilde{\mathbf{x}}_i^\top \Omega \tilde{\mathbf{x}}_j
\tag{3}
$$

in which $\mathbf{x}_i$ is the destination token and $\mathbf{x}_j$ is the source token of the attention computation. To enforce masked self-attention, $A'_{ij}$ is set to $-\infty$ for $i < j$. Attention scores are then normalized, for each destination (corresponding to a row in $A'$), yielding *attention weights* $A = \text{Softmax}(A'/\sqrt{r})$, in which the Softmax operation is performed for each row of $A'/\sqrt{r}$. The resulting attention weight $A_{ij}$ is the amount of attention that destination $i$ is placing on source $j$.

## 4 CIRCUIT TRACING

### 4.1 APPROACH

The approach used in this paper to trace circuits starts by decomposing the attention score $A'$ in terms of the SVD of $\Omega$. The matrix $\Omega$ has size $(d+1) \times (d+1)$, but due to its construction it has maximum rank $r$. We therefore work with the SVD of $\Omega = U\Sigma V^\top$ in which $U \in \mathbb{R}^{(d+1)\times r}$, $V \in \mathbb{R}^{(d+1)\times r}$ and $\Sigma \in \mathbb{R}^{r\times r}$. $U$ and $V$ are orthonormal matrices with $U^\top U = I$ and $V^\top V = I$, and $\Sigma = \text{diag}(\sigma_0, \sigma_1, \dots, \sigma_{r-1})$ with $\sigma_0 \geq \sigma_1 \geq \cdots \geq \sigma_{r-1} \geq 0$. Important to our work is that the SVD of $\Omega$ can equivalently be written as

$$
\Omega = \sum_{k=0}^{r-1} \mathbf{u}_k \sigma_k \mathbf{v}_k^\top = \sum_{k=0}^{r-1} D_k
\tag{4}
$$

in which $\{\mathbf{u}_k\}$ and $\{\mathbf{v}_k\}$ are orthonormal sets and each term in the sum is a rank-1 matrix having Frobenius norm $\sigma_k$. We refer to each term $D_k$ as an *orthogonal slice* of $\Omega$, since we have $D_k^\top D_j = D_kD_j^\top = 0$ whenever $k \neq j$.

The following hypothesis drives our approach:

**Hypothesis (Sparse Decomposition)** When an attention head performs a task that requires detecting components in a pair of low-dimensional subspaces in its inputs $\mathbf{x}_i$ and $\mathbf{x}_j$, and its inputs have significant components in those subspaces, it will show large values of $\mathbf{x}_i^\top \mathbf{u}_k \sigma_k \mathbf{v}_k^\top \mathbf{x}_j$ for a distinct subset of values of $k$.

In the remainder of this paper we show a variety of evidence that is consistent with the sparse decomposition hypothesis in the case of GPT-2 small. In §6 we will discuss reasons why this phenomenon may arise.

When the sparse decomposition hypothesis holds, we can approximate the score computed by the attention head as follows:

$$
A'_{ij} \approx \sum_{k \in S_{ij}} \mathbf{x}_i^\top \mathbf{u}_k \sigma_k \mathbf{v}_k^\top \mathbf{x}_j = \sum_{k \in S_{ij}} \mathbf{x}_i^\top D_k \mathbf{x}_j
\tag{5}
$$

where the number of terms in the sum (i.e., $|S_{ij}|$) is small.

We use $S_{ij}$ to denote the subset of values of $k$ for the token pair $(i, j)$. Besides being specific to an attention head and token pair, $S$ also depends on the task. In this paper we do not define 'task' precisely; in what follows, we study only a limited set of attention head functions that are performed when generating outputs for IOI prompts in GPT-2 small. We leave the association between $S$ and precisely-defined tasks as a fascinating direction for future work.

Our tracing strategy constructs $A'$ using (5), i.e., using $S_{ij}$ in place of all of the singular vectors of $\Omega$. This is akin to *denoising* in signal processing. When a signal has an approximately-sparse representation in a particular orthonormal basis (e.g., the Fourier basis or a wavelet basis) then removing the signal components that correspond to small coefficients is useful to suppress noise. Likewise, we find that in the orthonormal bases provided by $U$ and

$V$, contributions of the inputs $\mathbf{x}_i$ and $\mathbf{x}_j$ are typically approximately-sparse. Hence removing the dimensions with contributions summing to zero allows us to identify low-dimensional components of the inputs that are responsible for most of the attention head's output (score). We illustrate the benefits of denoising $A'$ in §5.2.

Hence, to fix $S_{ij}$, we consider the individual contributions made by each orthogonal slice to $A'_{ij}$: $\{\mathbf{x}_i^\top D_k \mathbf{x}_j\}_{k=0}^{r-1}$. Empirically we typically find a few large, positive terms and many others that may be positive or negative. We seek to separate terms into 'signal' and 'noise.' To do so we adopt a simple heuristic, treating noise terms as a set that, in sum, has little or no effect on the attention head score. Accordingly, we define the noise terms to be the largest set of terms whose sum is less than or equal to zero. The indices of the remaining terms constitute $S_{ij}$. Terms denoted by $S_{ij}$ are strictly positive, and are the largest positive terms. Typically the number of those terms, ie, $|S_{ij}|$, is 20 or less, often just 2 or 3. We refer to this condition where $|S_{ij}|$ is small as the *sparse decomposition* of attention head scores in terms of the orthogonal slices of $\Omega$.

Given $S_{ij}$, we can decompose model residuals into 'signal' and 'noise' in terms of their impact on $A'_{ij}$. Define subspaces $\mathcal{U} = \text{Span}\{\mathbf{u}_k \,|\, k \in S_{ij}\}$ and $\mathcal{V} = \text{Span}\{\mathbf{v}_k \,|\, k \in S_{ij}\}$ and associated projectors $P_{\mathcal{U}}$ and $P_{\mathcal{V}}$. The denoising step separates the inputs $\mathbf{x}_i$ and $\mathbf{x}_j$ into:

$$\tilde{\mathbf{s}}_i = P_{\mathcal{U}}\tilde{\mathbf{x}}_i, \quad \tilde{\mathbf{z}}_i = P_{\mathcal{U}^\perp}\tilde{\mathbf{x}}_i, \quad \tilde{\mathbf{s}}_j = P_{\mathcal{V}}\tilde{\mathbf{x}}_j, \quad \tilde{\mathbf{z}}_j = P_{\mathcal{V}^\perp}\tilde{\mathbf{x}}_j, \tag{6}$$

where $P_{\mathcal{U}^\perp} = I - P_{\mathcal{U}}$ and $P_{\mathcal{V}^\perp} = I - P_{\mathcal{V}}$. Then we have $\mathbf{x}_i = \mathbf{s}_i + \mathbf{z}_i, \mathbf{x}_j = \mathbf{s}_j + \mathbf{z}_j$, and

$$\tilde{\mathbf{s}}_i^\top \Omega \tilde{\mathbf{s}}_j \approx A'_{ij} \quad \text{and} \quad \tilde{\mathbf{z}}_i^\top \Omega \tilde{\mathbf{z}}_j \approx 0,$$

where $|S_{ij}|$ is as small as possible.[1] Intuitively, we interpret a signal $\mathbf{s}$ to approximately represent a feature that is used for communication between attention heads; we present evidence in support of this interpretation below.

## 4.2 Singular Vector Tracing

We use this framework to trace circuits in GPT-2 small as follows. A prompt corresponding to the IOI task is input to GPT-2. Consider the $a$-th attention head at layer $\ell$, generating attention score $A'_{ij}$ for source token $j$ and destination token $i$. Then $S_{ij}^{\ell a}$ defines a set of orthogonal slices that the attention head $(\ell, a)$ is using. Hence we can approximately construct $A'_{ij}$ as in (5), where we are using the SVD of $\Omega^{\ell a}$.

We will trace circuits causally with respect to the singular vectors of each attention head. For attention head $(\ell, a)$ generating output on tokens $(i, j)$, we identify the subspaces $\mathcal{U}$ and $\mathcal{V}$. We then look at each attention head 'upstream' of $(\ell, a)$, to determine how much each contributes in the subspace $\mathcal{U}$ to destination token $\mathbf{x}_i$ and in the subspace $\mathcal{V}$ to source token $\mathbf{x}_j$. Specifically, for attention head $(l, b)$ with $l < \ell$, denote the output that it adds to token $i$ as $\mathbf{o}_i^{lb}$ (likewise for $j$). We then compute

$$c_{i,ij}^{\ell a,lb} = \sum_{k \in S_{ij}^{\ell a}} \sqrt{\sigma_k^{\ell a}}\mathbf{u}_k^{\ell a\top}\mathbf{o}_i^{lb} \quad \text{and} \quad c_{j,ij}^{\ell a,lb} = \sum_{k \in S_{ij}^{\ell a}} \sqrt{\sigma_k^{\ell a}}\mathbf{v}_k^{\ell a\top}\mathbf{o}_j^{lb} \tag{7}$$

Conceptually, $c_{i,ij}^{\ell a,lb}$ is an estimate of how much attention head $(l, b)$, by writing to token $\mathbf{x}_i$, has changed the output (score) of attention head $(\ell, a)$ on token pair $(i, j)$. We use $\sqrt{\sigma_k}$ to incorporate the magnitude of each singular vector's contribution to the attention head output, dividing the contribution equally between the source and destination tokens. We refer to $c_{i,ij}^{\ell a,lb}$ as the *contribution* of attention head $(l, b)$ to attention head $(\ell, a)$ on token pair $(i, j)$ through token $\mathbf{x}_i$.

## 4.3 Experiments

To demonstrate the utility of singular vector tracing we apply it to a specific setting: the behavior of GPT-2 small when performing indirect object identification (IOI). The IOI problem was introduced in (Wang et al., 2023), and that paper identified circuits that GPT-2 uses to perform the IOI task. As described in that paper: *In IOI, sentences such as "When Mary and John went to the store, John gave a drink to" should be completed with "Mary."* To be successful, the model must identify the indirect object (IO, 'Mary') and distinguish it from the subject (S, 'John') in a prompt that mentions both. Thus, a succinct measure of model performance can be obtained by comparing the output logits of the IO and S tokens. The authors identified a collection of attention heads, and the token positions they attend to, that together perform the IOI task. We refer the reader to (Wang et al., 2023) for additional details.

---

[1] To go from $\tilde{\mathbf{x}}$ to $\mathbf{x}$ we simply drop the last component; this is explained in the Appendix.

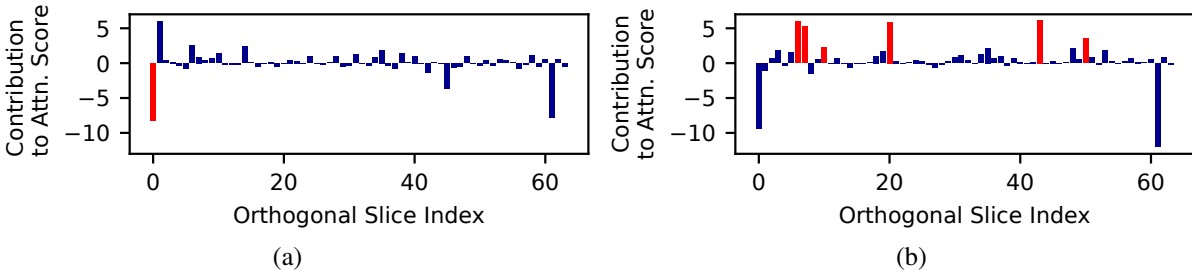

Figure 1: Orthogonal slice contributions to attention score (a): $A'_{ij} < 0$; (b) $A'_{ij} > 0$.

The IOI dataset that we use consists of 256 example prompts with 106 different names. We are using the same 15 templates used in (Wang et al., 2023), with two patterns ('ABBA' and 'BABA'), which refers to the order of the names appearing in the sentence (the IO name is A, and the S name is B). Prompt sizes range from 14 to 20 tokens.

Singular vector tracing as described in §4.2 can be applied at every attention head with respect to every token pair. However in our experiments we limit our analysis to cases where the attention head is primarily attending to a single source token for a given destination token. This strategy is similar to analyses in prior work (Wang et al., 2023; Conmy et al., 2023; Ferrando & Voita, 2024); it reduces complexity in the tracing analysis, but could be relaxed in future work. Specifically we say that an attention head is 'firing' if it places more than 50% weight on a particular source token for any given destination. This rule implies that a head can only 'fire' on one source token for each destination token. In general, we only trace upstream from attention heads that are firing on specific token pairs.

Note that interpreting (7) as giving the actual magnitude of the change in downstream attention score overlooks processing that may affect the signal in between the upstream head and the downstream head. Previous work has shown that downstream processing can compensate for upstream ablations (McGrath et al., 2023) and that some layers may remove features added by previous layers (Rushing & Nanda, 2024). We show results in §5.4 that confirm the direct causality of signals on downstream attention head outputs, but also illustrate that downstream processing can at times have a noticeable effect on model performance after signal interventions. Further, (7) ignores the impact of the layer norm. We account for the effect of the layer norm using three techniques: weights and biases are folded into the downstream affine transformations, output matrices are zero centered,[2] and the scaling applied to each token is factored into the contribution calculation. More details on tracing are provided in the Appendix.

## 5 RESULTS

### 5.1 CHARACTERIZING ATTENTION HEAD BEHAVIOR VIA SVD

Throughout this and the next section, we use as our examples attention heads that figured prominently in the results of (Wang et al., 2023). This aids interpretation and ensures we are paying attention to important components of the model.

We start by demonstrating the approximately-sparse nature of attention scores when decomposed via SVD. Figure 1 shows typical cases. Each plot in the figure shows results for attention head $(8, 6)$ and a single source token, destination token, and prompt. The 64 contributions made by each orthogonal slice to the attention score are shown. Red bars correspond to sets $S_{ij}$; the sum of the red bars is approximately equal to the sum of all bars, which is the attention score for this head on these inputs. Figure 1(a) corresponds to the case in which the attention score is negative (and so the attention weight would be nearly zero); Figure 1(b) corresponds to a case in which the attention score is positive (28.1), and the attention weight is large ($\approx 0.83$). Note that orthogonal slices are shown in order of decreasing singular value; the effect shown is not due to a low-rank property of $\Omega$ itself. Rather, the effect shown corresponds to sparse construction of the attention score when the *inputs* are encoded in the bases given by the SVD of $\Omega$.

We see considerable evidence in our experiments that the orthogonal slices used by an attention head are similar to each other when the attention head is firing. Figure 2 shows examples of the $S_{ij}$ sets for four attention heads: $(3, 0)$, $(4, 11)$, $(8, 6)$, and $(9, 9)$ across 256 prompt inputs. Furthermore, the nature of these attention head's functions are evident in the sets of slices that they use. In the case of $(8, 6)$ (an S-inhibition head), there is a set of about 6 slices that are consistently used; these are the same as the red bars in Figure 1(b). Attention head $(9, 9)$ (a name mover head) uses a larger set, but there is still clearly a specific set of slices that frequently appear. Attention head $(3, 0)$ (a duplicate-token head) uses a broad set of slices; this is consistent with its need to look at all the token's dimensions,

---

[2]These operations are provided by the TransformerLens library; see (Nanda).

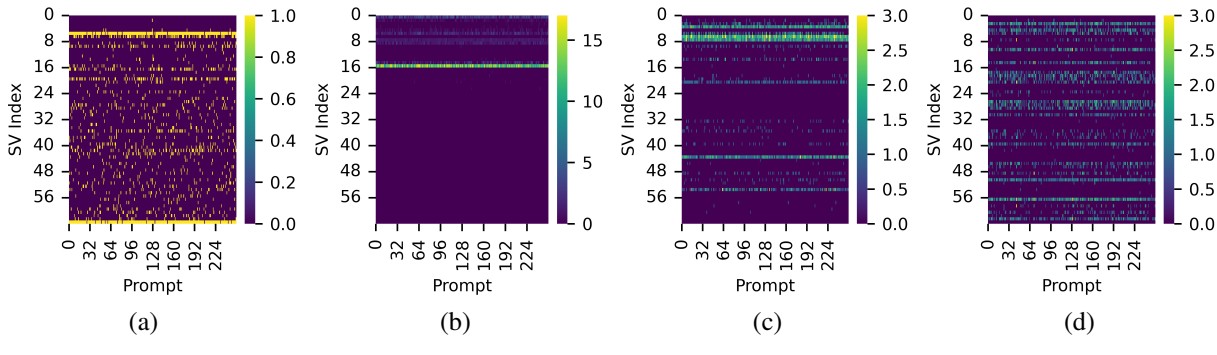

Figure 2: Orthogonal slices used when head is firing: (a) AH (3, 0); (b) AH (4, 11); (c) AH (8, 6); (d) AH (9, 9).

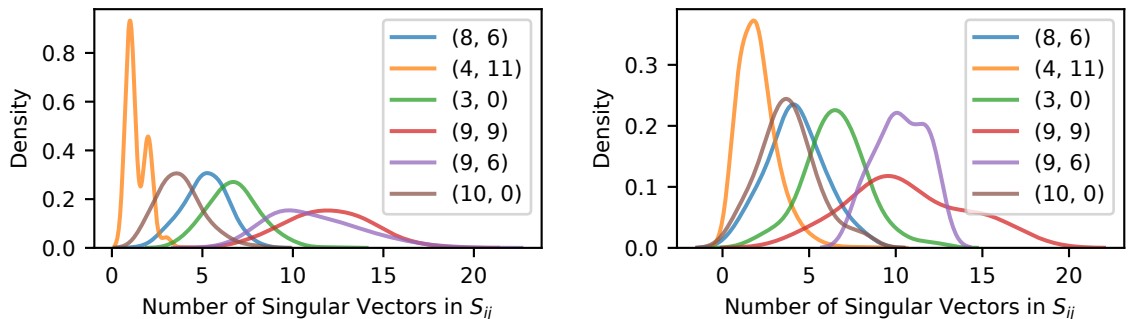

Figure 3: Number of slices used when firing. (left) IOI dataset (right) non-specific dataset.

since its role is to detect identical tokens. And attention head (4, 11) (a previous token head) primarily uses a single slice; this also is consistent with its role of detecting adjacent tokens, which appears to only require detecting a feature in a one-dimensional subspace. Finally, we show in the Appendix corresponding plots for when these attention heads are *not* firing; there too, a consistent set of slices is used, but the slices are completely different from those used when the head is firing.

The sparsity of the contributions of the slices of $\Omega$ to attention scores is summarized in Figure 3. In Figure 3(a) we show the distribution of the number of orthogonal slices used, ie, $|S_{ij}|$, across all the prompts in our dataset. The figure shows that the number of slices used is consistently small, much smaller than the number of available dimensions (64). We also ask whether the sparsity observed is an artifact of the input data used in our experiments. To assess this, we run GPT-2 on a dataset consisting of non-specific inputs, having no relationship to the IOI task. We use the first 256 elements of The Pile (Nanda, 2024) selecting the first 21 tokens from each element to match the IOI dataset size. The corresponding plot is shown in Figure 3(b). This comparison suggests that sparse decompositions of attention scores is not limited to the IOI setting.

## 5.2 Do We Need Singular Vectors?

Next we show that it is possible to leverage the sparsity of attention score decomposition. To illustrate this, we ask the following question: given a token that is input to a particular attention head, what similarity does it show to the outputs of upstream attention heads? In other words, could we trace some part of a circuit by simply looking upstream to see who has 'contributed' to the token?

We take as our examples heads (9, 9) and (10, 0), which are name mover heads. The authors in (Wang et al., 2023) find that functionally important contributors to the 'end' tokens of these heads are (7, 3), (7, 9), (8, 6), and (8, 10). The heatmaps in Figure 4 show contribution scores (computed via (7)) for two cases: the case where the signal is taken to be the entire residual $\mathbf{x}$, and the case where the signal $\mathbf{s}$ is taken to be just its low-dimensional component as described in §4.1.

The figure shows the strong filtering and correcting effect that results from exploiting the sparsity of attention decomposition. Figures 4(a) and (c) show that if we simply ask how much each upstream head contributes to the attention score of the downstream head, we get a very noisy answer with two problems. First, a large set of attention heads have

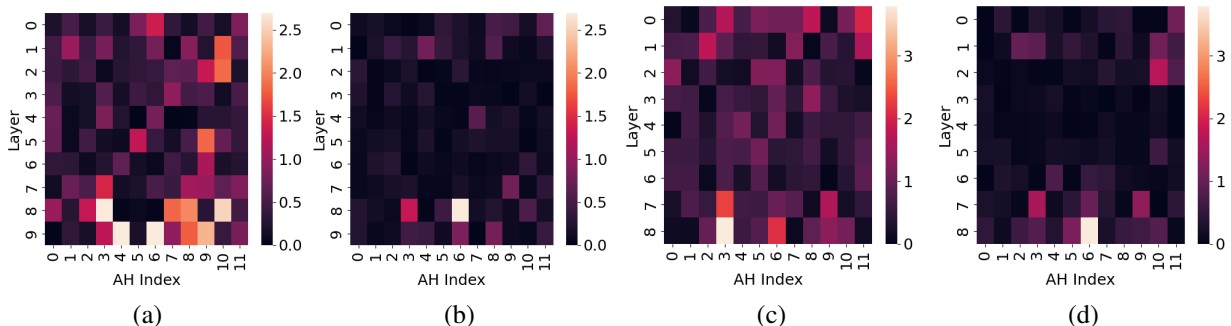

Figure 4: Filtering effect of orthogonal slices. Upstream contributions to (a) (10, 0), 'end' token, all slices of $\Omega$; (b) minimal set of slices of $\Omega$; (c) (9,9), 'end' token, all slices of $\Omega$; (d) minimal set of slices of $\Omega$.

high scores; and second, the known functional relationships from (Wang et al., 2023) are not evident. On the other hand, when we focus on Figures 4(b) and (d), we see considerable noise suppression; many heads in middle layers of the model are no longer shown as being significant. Furthermore, we see that one attention head (8, 6) stands out, and it is one of those previously identified as functionally important. And in the case of Figure 4(d), we see that other attention heads with known functional relationship (7, 3) and (7, 9), are also highlighted. In §5.3 we will show many other attention head pairs with functional relationships from (Wang et al., 2023) that are recovered using the singular vector tracing strategy. In the Appendix we show a hypothetical network trace performed without using singular vectors; the resulting trace is not usable and shows little evidence of known functional relationships in the IOI circuit.

**Interpreting Signals.** Detailed investigation of the interpretability of signals is beyond the scope of this paper. However we observe that in some cases signals show interpretability. As an example, we consider the name mover attention head (9, 9). For each token in the input to layer 9, we compute the magnitude of its residual in the $\mathcal{V}$ subspace of (9, 9). This measures the strength of the signal that (9, 9) uses to identify the IO token. We find that this measure cleanly separates the names in our data from the non-names. Details are in the Appendix.

5.3 SINGULAR VECTOR TRACE OF GPT-2 ON IOI

Next we construct a singular vector trace using the concepts from §4.2. We start at a particular attention head and token pair. If the head is firing on the token pair (as discussed in §4.3) we obtain the subspaces $\mathcal{U}$ and $\mathcal{V}$ and associated projectors $P_\mathcal{U}$ and $P_\mathcal{V}$. We then look at each upstream attention head's output, and separate from it the signal it contains for the downstream head. The properly adjusted magnitude of this signal constitutes the upstream head's contribution to the downstream head's attention score via the corresponding token, as computed via (7).

Empirically we find that the contributions from most upstream heads are small, with only a few upstream heads making large contributions to the downstream attention score. We filter out the small contributions, which are unlikely to have significant impact on model performance. To do this for a given downstream firing, we adopt the simple rule of choosing the smallest set of upstream heads whose contributions sum to at least 70% of the sum of all contributions.

Using this rule, for each token we identify the upstream heads with significant contribution through that token. An edge in the resulting trace graph is defined by the upstream head, the downstream head, the two tokens on which the downstream head is firing, and the choice of which token is being written into by the upstream head. For each upstream head we then ask whether it is firing on that token as a destination. If so, the process repeats from that head and token pair. The process is presented in detail as Algorithm 1 in the Appendix.

We ran singular vector tracing on GPT-2 small using 256 prompts from the IOI dataset described in §4.3. We started the trace at the three name mover heads (9, 6), (9, 9) and (10, 0), as they were identified as having direct effect on model performance in (Wang et al., 2023). The resulting trace is shown in Figure 5; we refer to the graph in the figure as $G$. There are two kinds of edges in $G$: communication edges from heads to tokens, and attention edges from tokens to heads. In the figure, blue edges are toward tokens that are source tokens downstream, and red edges likewise are destinations. The width of the edge is the accumulated contribution of the edge over the 256 prompts. Darker nodes fired more often in our traces, and we have placed green borders around nodes that appeared in (Wang et al., 2023). Edges that appear very few times (less than 65 times over the 256 prompts) are omitted.

We make a number of observations. First, the trace shows broad agreement with results in (Wang et al., 2023), although not all nodes in that paper appear in our graph; this may reflect differences of effect size, or differences between the input datasets. The trace shows agreement with previous results on head-to-head connections and also on the tokens

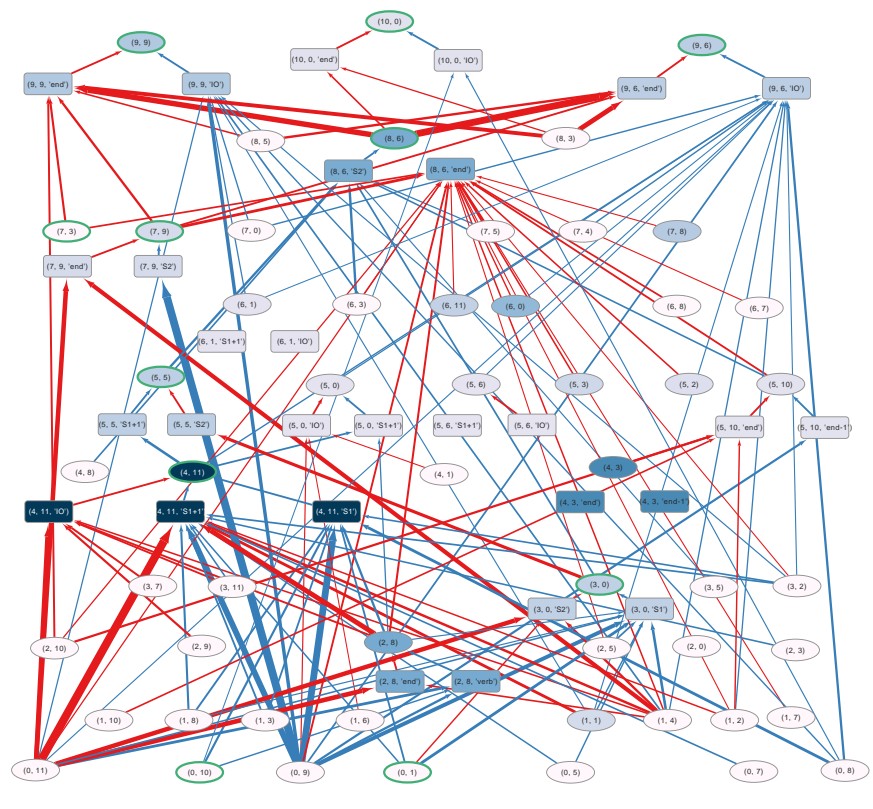

Figure 5: Traced Network, 256 Prompts. Heads are ovals, tokens are boxes.

through which the communication is effected. Note that in the appendix we show that further filtering this graph to just its most frequent edges yields a set of attention heads in agreement with (Wang et al., 2023) to a precision of 0.52 and recall of 0.69.

The trace also goes beyond previous results in a number of ways. Whereas in previous work, the upstream contributors to name mover source tokens were not identified, this trace shows where important features for the that (IO) token are added and that this happens very early in the model's processing. Previous work also proposed that redundant paths were present in GPT-2's processing for the IOI task; this trace confirms their existence and elucidates the nature of their interconnection pattern. For example, there is distinct lattice structure among nodes at layers 7, 8, and 9. (In the Appendix we isolate this lattice structure for better inspection.) We also identify highly active heads that were not discussed in previous work, including (2, 8) which attends to the ditransitive verb of the prompt and feeds into (4, 11) (a previous token head) as well as (8, 6) (a S-inhibition head); and (4, 3) which attends to the last token (which is always a preposition in our prompts) and its predecessor.

## 5.4 VALIDATION

We adopt a variety of strategies to validate the graph in Figure 5. To demonstrate the causal effect of each edge's communication on model performance, we intervene on individual edges; and to demonstrate that the structure of the graph itself is functionally significant, we intervene on various collections of edges simultaneously.

**Edge Validation.** A communication edge in $G$ represents the contribution $c_{ij}^{\ell a, lb}$, which is a measure of the amount that head $(l, b)$ would change the attention score $A'_{ij}$ of head $(\ell, a)$ via $\mathbf{s}_{ij}^{\ell a, lb}$ if there were no downstream modifications. Validating an edge $(l, b) \rightarrow (\ell, a)$ in $G$ involves intervening in the output of the upstream attention head $(l, b)$ by modifying the signals (as defined in (6)) used by the downstream attention head $(\ell, a)$. We define two types of interventions: *global* interventions, and *local* interventions. In a global intervention, we simply modify the signal in the output of the upstream attention head; in a local intervention, we modify the signal *only* at the input to the downstream attention head. Global interventions have the potential to directly affect all downstream heads, while local interventions are limited in their direct effect to only the downstream head.

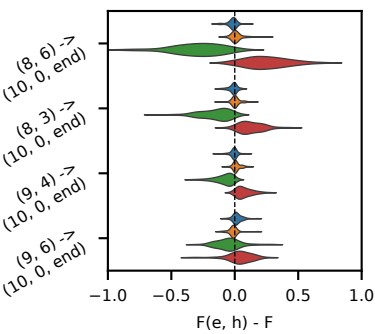 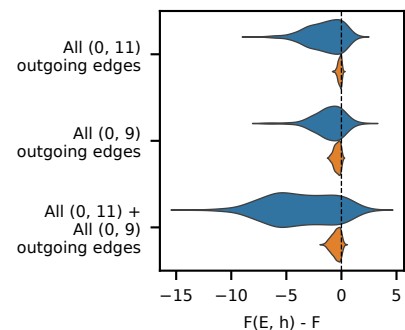

Figure 6: Single-Edge Interventions: (left) Global (right) Local. Green: Ablation; Red: Boosting; Blue: Random Ablating; Orange: Random Boosting.

Figure 7: Multi-Edge Ablation. Blue: Global; Orange: Local.

Specifically, we define $\Delta$ as the projection of the upstream head's output onto the associated subspace defined by $S_{ij}^{\ell a}$. An edge *ablation* consists of subtracting $\Delta$ from the residual; and edge *boosting* consists of adding $\Delta$ to the residual. Additionally, for comparison purposes we define $\Delta_{\text{random}}$ which consists of constructing the projection of the residual using a set of $(\ell, a)$'s singular vectors chosen at random from those not in $S_{ij}^{\ell a}$; the size of the chosen set is the same as $|S_{ij}^{\ell a}|$. Performing the same interventions using $\Delta_{\text{random}}$ allows us to assess whether intervening in the subspaces $\mathcal{U}$ and $\mathcal{V}$ is more effective than a random strategy. Implementation details of these interventions are in the Appendix.

We measure the effects of interventions using the same metric as (Wang et al., 2023). Let $F(X)$ be the logit difference between the IO and S tokens when the model is run on input $X$. Then define $F(X, e, \mathbf{h}) = F(X \mid \text{do}(\mathbf{x} = \mathbf{x} + \mathbf{h}))$ for a particular intervention $\mathbf{h}$ on a particular edge $e$ corresponding to residual $\mathbf{x}$. A negative value of $F(X, e, \mathbf{h}) - F(X)$ indicates that the model's performance on the IOI task has gotten worse; while a positive value of $F(X, e, \mathbf{h}) - F(X)$ indicates that the model's performance has improved.

Our results show, with some interesting exceptions, that almost all of the edges from $G$ that we test cause model performance to degrade when ablated. Importantly, we also show that model performance shows corresponding *increases* when an edge is boosted. Generally speaking, intervening on an edge with higher weight (shown as thicker) in Figure 5 has stronger impact on model performance. We conclude that tracing via singular vectors generally identifies functionally causal communication paths for IOI in GPT-2.

It is important to note that the magnitudes of the interventions performed here are small. We are intervening in a subspace of dimension less than 20 (typically) on a vector with 768 dimensions. To illustrate, we find that the cosine similarity between residuals before and after a local intervention in our experiments is generally higher than 0.999; and the relative change in vector norm before and after intervention is typically less than 1%. Details are in the Appendix.

As an example of our results, we show in Figure 6 interventions on edges into (10, 0, end). Both local and global interventions show causal impact on model performance, with more significant edges having larger impact. We note that in a number of cases (shown in the Appendix) local interventions can have greater impact than global interventions, presumably due to downstream modification of signals by other components; we also show cases where global interventions have greater impact than local interventions, presumably because signals are being shared between multiple communication paths in the model.

**Structural Validation.** The results in (Wang et al., 2023) suggest that there are alternative paths that can be used by the model to perform the IOI task. An advantage of the trace shown in Figure 5 is that those alternate paths are made explicit. To demonstrate that the structure of $G$ is informative, we show that intervening on parallel edges is frequently additive, while intervening on serial edges is frequently not.

Figure 7 shows a multi-edge intervention in a large set of edges on parallel paths: all outgoing edges from attention heads (0, 11) and (0, 9). Global interventions of these edges have significant impact in the performance of the model, and this effect is amplified when we intervene on both (0, 11) and (0, 9) at the same time, showing intervention on parallel paths that is additive. On the other hand, local intervention in the downstream nodes of these edges has a smaller effect, suggesting that the model has redundant paths for the same task (eg, backup name movers as discussed in (Wang et al., 2023)). Results showing intervention on serial edges are in the Appendix.

# 6 DISCUSSION

**Possible Mechanisms.** The results in §§5.1 and 5.2 support the sparse decomposition hypothesis. In this section we discuss some possible mechanisms behind this phenomenon. First, a motivation for decomposing attention matrices using SVD comes from the following fact:

**Lemma 1** Given vectors $\mathbf{x}$ and $\mathbf{y}$, among all rank-1 matrices having unit Frobenius norm, the matrix $D$ that maximizes $\mathbf{x}^\top D \mathbf{y}$ is $D = \frac{\mathbf{x}}{\|\mathbf{x}\|} \frac{\mathbf{y}^\top}{\|\mathbf{y}\|}$.

The proof is straightforward and provided in the Appendix. This suggests that model training could have the following effect. If an attention head needs to attend to particular vectors $\mathbf{x}$ and $\mathbf{y}$, it needs to output a large value for $\tilde{\mathbf{x}}^\top \Omega \tilde{\mathbf{y}}$. In that case, model training could result in construction of $\Omega$ in which one term of the SVD, say $\mathbf{u}_k \sigma_k \mathbf{v}_k^\top$, has $\mathbf{u}_k \approx \tilde{\mathbf{x}}/\|\tilde{\mathbf{x}}\|$, $\mathbf{v}_k \approx \tilde{\mathbf{y}}/\|\tilde{\mathbf{y}}\|$, and with $\sigma_k$ reflecting the importance of this term in the overall computation of the attention score. As an illustration, we note that the results in §5.2 and the Appendix show that in GPT-2 the singular vectors of some $\Omega$ matrices are correlated with word features that are relevant for the IOI task.

Consider a hypothetical case in which the sets of vectors to which the attention head needs to attend, say $\{\tilde{\mathbf{x}}_i\}$ and $\{\mathbf{y}_i\}$, happen to each form orthogonal sets. Then to achieve maximum discrimination power in distinguishing corresponding pairs, the singular vectors of $\Omega$, that is $\{\mathbf{u}_i\}$ and $\{\mathbf{v}_i\}$ respectively, should be aligned with the corresponding vectors in $\{\tilde{\mathbf{x}}_i\}$ and $\{\tilde{\mathbf{y}}_i\}$. To move from vectors to features, we refer to the *linear representation hypothesis* (Mikolov et al., 2013; Gurnee & Tegmark, 2024; Park et al., 2023) which suggests that concepts, including high-level concepts, are often represented linearly in the model.

To move closer to realistic cases, we note that the authors in (Elhage et al., 2022) make observations, based on experimental evidence and geometric considerations, about features constructed by neural models. They argue that models will tend to represent correlated feature sets in a manner such that, considered in isolation, the sets are *nearly* orthogonal. They term this the use of "local, almost-orthogonal bases." In our case, if an attention head is attending to vector sets $\{\tilde{\mathbf{x}}_i\}$ and $\{\tilde{\mathbf{y}}_i\}$ that are important when performing a specific task, then we may hypothesize that training will construct the sets to be "nearly-orthogonal," meaning that cosine similarities among the vectors in each set would typically be small. In this case, the resulting sets of singular vectors $\{\mathbf{u}_k\}$, $\{\mathbf{v}_k\}$ are more likely to *sparsely encode* the $\{\tilde{\mathbf{x}}_i\}$ and $\{\tilde{\mathbf{y}}_i\}$ than exist in one-to-one correspondence.

Given the above argument, for cases where the linear representation hypothesis holds, we expect that an attention head is testing for a pair of low-dimensional subspaces in the inputs $\mathbf{x}_i$ and $\mathbf{x}_j$. In that case, we expect that subsets of the singular vectors of $\Omega$ will be constructed during training so as to 'match' those subspaces. In this context, a sparse encoding allows the attention head to attend to more than $r$ different subspaces, expanding the number of concepts that the attention head can recognize. In the Appendix we discuss situations where the sparse decomposition hypothesis may not hold.

**Limitations and Future Work.** There are a number of limitations of our study and directions for future work. First, the method as used here does not explore alternative pathways that may affect model output (Makelov et al., 2023). It also does not directly assess adaptive computations in the model (Marks et al., 2024; Rushing & Nanda, 2024) nor does it construct minimal circuits in the sense of (Wang et al., 2023). However, we believe that it offers an alternative toolbox that can be extended to help investigate those issues, in part by exposing the signals passing between specific pairs of attention heads. And as in (Wang et al., 2023), we focus on understanding the interaction between attention heads. We believe that extending our framework to include the contributions of MLPs is an important direction for future work.

Further, we have not explored in depth the nature of the signals themselves. Indirect evidence, such as the difference in effect between local and global ablations, suggests that there are some similarities between signals used on different edges of the network. Exploration of the nature of signals and their relationships is an intriguing direction for future work.

**Conclusions.** Transformer-based models are largely considered closed boxes whose internals are difficult to interpret in domain-level terms (Alishahi et al., 2019). By helping elucidate the circuits used by a language model in performing a given task, we hope to improve our ability to assess, validate, control, and improve model functions. In this paper we draw attention to a powerful tool for analyzing circuits: the fact that attention scores are typically sparsely constructed. This effect leads directly to the ability to identify signals used by attention heads to effect inter-head communication. We show that these low-dimensional signals have causal effect on attention head computations, and generally on the ability of GPT-2 to perform the IOI task. We believe that further exploration of signals holds promise for even deeper understanding of the internals of language models.

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

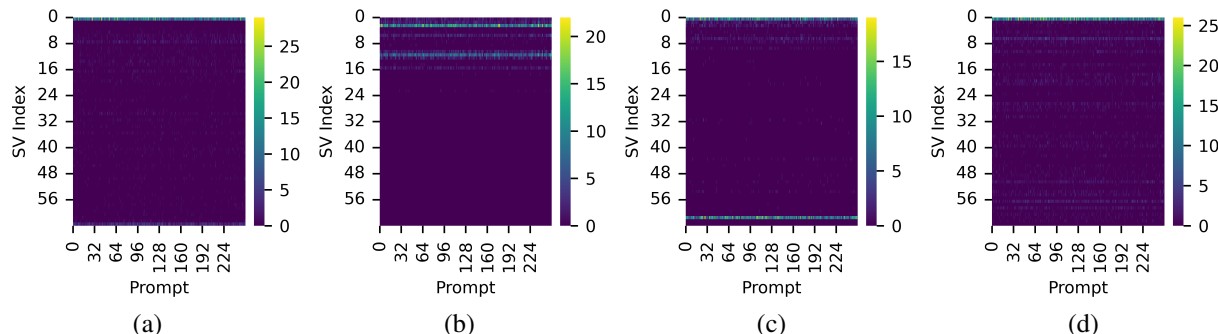

Figure 8: Orthogonal slices used when AH is *not* firing, 256 prompts. (a) AH (3, 0); (b) AH (4, 11); (c) AH (8, 6); (d) AH 9, 9). Compare to Figure 2.

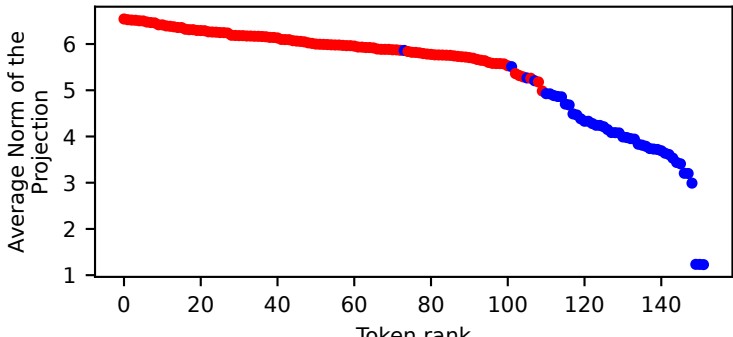

Figure 9: Average Magnitude of the (9, 9) $\mathcal{V}$ Space Signal in Each Token. Tokens corresponding to names are in red.

# A APPENDIX

**Handling Homogeneous Coordinates.** The $\Omega$ matrix as defined in (2) expects inputs in homogeneous coordinates in which the last component of the input vector is 1. This allows the bilinear form (3) to incorporate the linear and constant terms in (1). When projecting a residual to isolate a signal component, we are only concerned with the dimensions represented in the residual. We are not concerned with the additional $(d + 1)$th dimension, as that dimension represents the linear and constant terms that will be added by the downstream attention head. Hence, to compute the projection of a residual $\mathbf{x}$ in the subspace associated with $P$, we first form $\tilde{\mathbf{x}}$ by extending $\mathbf{x}$ with a $(d + 1)$th component having value 1, then compute $\tilde{\mathbf{s}} = P\tilde{\mathbf{x}}$, then drop the $(d + 1)$th component of $\tilde{\mathbf{s}}$ leaving $\mathbf{s}$.

**Slices Used When a Head is *Not* Firing.** In Figure 8 we show analogous plots to Figure 2, except that we choose cases where the attention heads are not firing. Interestingly, a consistent and small set of orthogonal slices is used in each head when it is not firing. However, comparing to Figure 2, we see that an entirely different set of orthogonal slices are in use when the head is not firing. We note that across our measurements we find a very small number of slices are responsible for an attention head's score when it is not firing.

**Interpretability of Signals.** As described in §5.2, we find that signals can show interpretability. As a simple illustration, we consider the (9, 9) attention head, a name mover. The projection of a token's residual into the $\mathcal{V}$ subspace of this attention head isolates the signal used by the head to recognize the IO token.

To make this a uniform measure we can apply across multiple tokens, we create a consensus estimate of a single subspace $\mathcal{V}$ across all firings of (9, 9). We do this by selecting only the orthogonal slices that appear in at least 100 firings. This yields 12 sets of singular vectors which we can use to construct $P_\mathcal{V}$. We then measure $\|P_\mathcal{V}\mathbf{x}\|$ for all token residuals $\mathbf{x}$ at the input of layer 9. The average of the norms for each token is plotted in Figure 9. Tokens are sorted by the magnitude of the signal $\|P_\mathcal{V}\mathbf{x}\|$ and colored according to whether they are names (in red) or not (in blue). We see that magnitude of the signal in $\mathbf{x}$ is a clear measure of whether that token is a name, ie, a potential IO.

**Tracing Details.** Here we discuss some of the fine points of our tracing method.

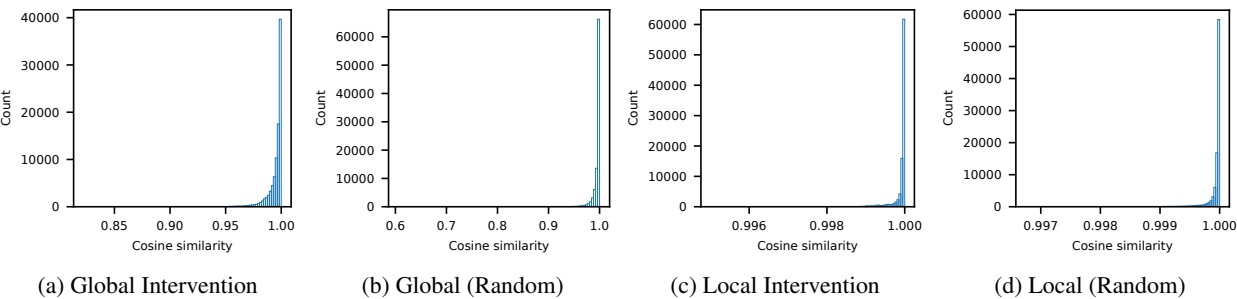

(a) Global Intervention     (b) Global (Random)     (c) Local Intervention     (d) Local (Random)

Figure 10: Distribution of cosine similarities between $\mathbf{x} + \mathbf{h}$ and $\mathbf{x}$ across single-edge interventions.

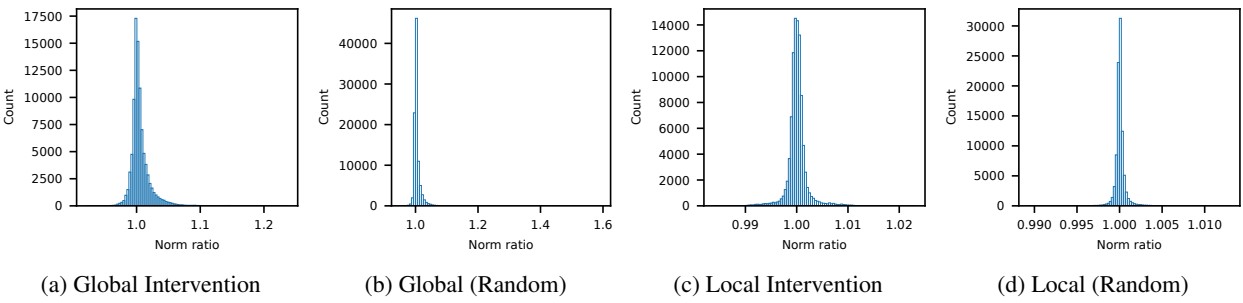

(a) Global Intervention     (b) Global (Random)     (c) Local Intervention     (d) Local (Random)

Figure 11: Norm ratio $\left( \frac{\|\mathbf{x}+\mathbf{h}\|}{\|\mathbf{x}\|} \right)$ distribution across single-edge interventions.

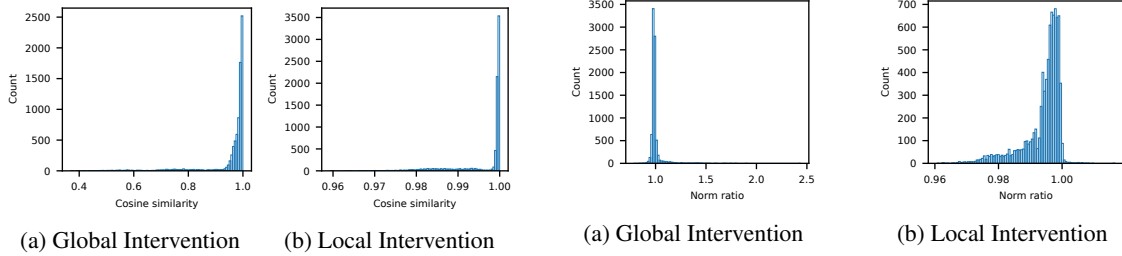

(a) Global Intervention     (b) Local Intervention          (a) Global Intervention     (b) Local Intervention

Figure 12: Distribution of cosine similarities between $\mathbf{x} + \mathbf{h}$ and $\mathbf{x}$ across multi-edge ablations.

Figure 13: Norm ratio $\left( \frac{\|\mathbf{x}+\mathbf{h}\|}{\|\mathbf{x}\|} \right)$ distribution across multi-edge ablations.

First, note that for each term in (5), we get the same value if we replace $\mathbf{u}_k, \mathbf{v}_k$ with $-\mathbf{u}_k, -\mathbf{v}_k$. For consistency of interpretation, we adopt the convention of defining $\mathbf{u}_k$ to lie in the direction that creates a positive inner product with $\mathbf{x}_i$. This determines the direction for $\mathbf{v}_k$ (and if $A'_{ij} > 0$, then $\mathbf{v}_k$ will lie in the direction of positive inner product with $\mathbf{x}_j$). The result is that we can treat the set of vectors $\{\mathbf{u}_k \mid k \in S^{\ell a}_{ij}\}$ as rays defining a cone of positive influence on $A'_{ij}$, and similarly for $\{\mathbf{v}_k \mid k \in S^{\ell a}_{ij}\}$.

As in (Wang et al., 2023), we focus on understanding the interaction between attention heads. We believe that extending our framework to include the contributions of MLPs is an important direction for future work. Furthermore, in the analyses in this paper we do not consider cases in which heads attend to the first token – that is, when attention head has a large value of $A_{ij}$ for $j = 0$. Because the attention weights for each target token form a probability distribution, when a destination token should not be meaningfully modified, attention heads normally put their weight on the first token. This role for token 0 has been noted in previous work (Nanda).[3]

Finally, as noted in §4.3, to properly attribute contributions of upstream heads to downstream inputs, we need to take into account the effect of the (downstream) layer norm. The layer norm operation can be decomposed into four steps: centering, normalizing, scaling, and translation. Centering, scaling, and translation are affine maps, which means that they can be folded into different parts of the model with mathematical equivalence. The TransformerLens library handles the centering step by setting each weight matrix that writes into the residual stream to have zero mean. Moreover, it folds the scaling and translation operations into the weights of the next downstream layer.[4] The result is that centering, scaling, and translation make changes to the matrices used to compute $\Omega$ as shown in (2). The remaining step is the normalizing step. This step does not change the direction of the residual; it only affects the magnitude of the contribution calculation (7). Since for any contribution calculation, we are considering a specific addition to the residual $\mathbf{o}_i$, we can simply scale its contribution by the same scaling factor used for the corresponding token $\mathbf{x}_i$ when it is input to the downstream layer.

**Algorithm for Singular Vector Tracing.** Singular vector tracing starts from a given head, token pair, and prompt and works upstream in the model to identify causal contributions to that attention head's output. If the upstream attention heads are themselves firing (attending mainly to a pair of tokens), then the process proceeds recursively for each token. We present singular vector tracing as Algorithm 1.

---

[3]GPT-2 does not use a 'bos' token.

[4]See `https://github.com/TransformerLensOrg/TransformerLens/blob/main/further_comments.md` for more details.

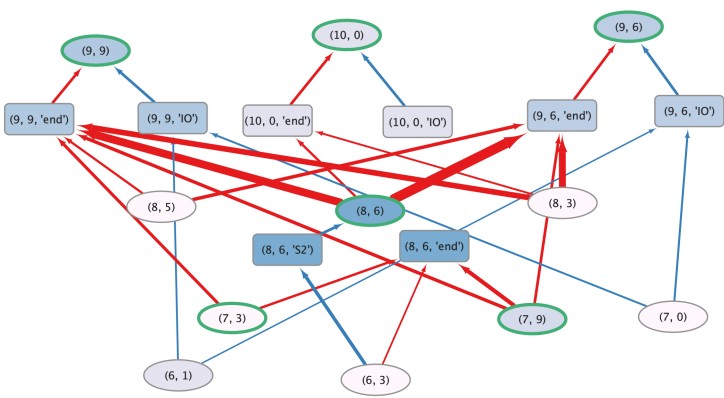

Figure 14: Top Layers of Traced Network, Showing Structure of Redundant Paths.

```
1  def is_not_firing(prompt, layer, ah_idx, dest_token, src_token):
2      if head (layer, ah_idx) on prompt tokens (dest_token, src_token) has attention weight < 0.5:
3          return True
4      else:
5          return False
6  def SVT(prompt, layer, ah_idx, dest_token, src_token):
7      edges = [ ]
       # see §4.3
8      if is_not_firing(prompt, layer, ah_idx, dest_token, src_token):
9          return edges
       # see §4.3
10     if src_token == 0:
11         return edges
       # See §4.2, Eqn (7)
       # For noise filtering, ignore small contributions as described in §5.3
12     src_contrib_ahs = upstream heads with significant contribution to src_token for (prompt, layer, ah_idx,
         dest_token, src_token)
13     dest_contrib_ahs = upstream heads with significant contribution to dest_token for (prompt, layer, ah_idx,
         dest_token, src_token)
14     for (upstream_layer, upstream_ah_idx) in src_contrib_ahs:
15         edges.append(layer, ah_idx, upstream_layer, upstream_ah_idx, src_token, dest_token, contrib, 's')
16         upstream_dest = src_token
17         for upstream_src in range(upstream_dest+1):
18             edges = edges + SVT(prompt, upstream_layer, upstream_ah_idx, upstream_dest, upstream_src)
19     for (upstream_layer, upstream_ah_idx) in dest_contrib_ahs:
20         edges.append(layer, ah_idx, upstream_layer, upstream_ah_idx, src_token, dest_token, contrib, 'd')
21         upstream_dest = dest_token
22         for upstream_src in range(upstream_dest+1):
23             edges = edges + SVT(prompt, upstream_layer, upstream_ah_idx, upstream_dest, upstream_src)
24     return edges
```

**Algorithm 1:** Python Pseudocode for Singular Vector Tracing.

**Alternative Paths in the IOI Circuit**  Previous work has noted that interventional studies are complicated by the presence of alternative causal paths in the IOI circuit. In Figure 14 we isolate nodes from the top layers of our network trace. The figure illustrates the lattice-like structure existing between the attention heads responsible for the higher-level processing in the circuit. The heads that take part in alternate pathways include the three name mover head (9, 6), (9, 9), and (10, 0), and the three S-inhibition heads (7, 3), (7, 9), and (8. 6).

**Skeleton of SVT Graph.**  In Figure 15 we show a version of singular vector trace shown in Figure 5. In this figure we have shown only edges that appear very frequently in our traces. Specifically, we show edges and (the nodes they connect to) that occur more than 170 times in our trace of 256 prompts. Because our trace strategy starts from (9, 6),

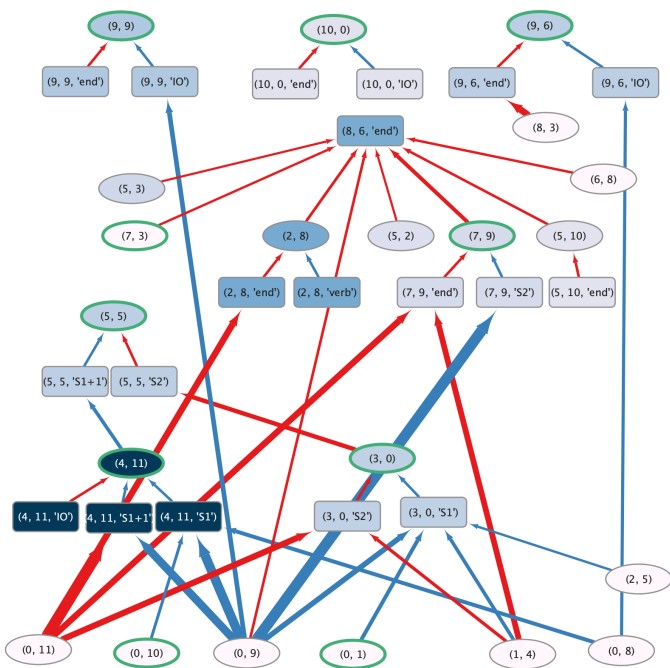

Figure 15: Skeleton of Traced Network. Edges filtered to 170 occurrences or more.

(9, 9), and (10, 0), only 16 of the heads identified in (Wang et al., 2023) can appear in the trace. The figure shows that this 'skeleton' contains 11 of the 16 possible attention heads previously identified, out of a total of 21 attention heads in the trace (precision $\approx 0.52$, recall $\approx 0.69$).

**Tracing Without Singular Vectors.** In §5.2 we show the noise suppression effect of filtering signals using the orthogonal slices of $\Omega$. That section shows that using all the orthogonal slices of $\Omega$, ie, simply looking at the residuals directly, without extracting their signals, is very noisy and leads to incorrect conclusions. As a further illustration, in Figure 16 we demonstrate a attempted circuit trace in which residuals are directly used for tracing, instead of using signals as in all other traces in this paper. The figure shows that the resulting trace is not useful. It does not contain most of the nodes that were identified as functionally important in (Wang et al., 2023). Further, it seems that contributions to source tokens are almost completely missed. The figure also shows that the most noisy and incorrect parts of the graph concern the longer-range connections between early and late layers of the model.

**Intervention Details.** Here we provide precise descriptions of the intervention strategies used in §5.4. Assume that, for a given instance, we are intervening in an edge $(l, b) \rightarrow (\ell, a)$ of type $t$ (either source or destination), with destination token $\tilde{\mathbf{x}}_i$, source token $\tilde{\mathbf{x}}_j$.

Global interventions are performed at the upstream head $(l, b)$ while local interventions are done at the downstream head $(\ell, a)$. We denote the $\tilde{\mathbf{x}}_m$ the token position that will be intervened. Specifically, if head $(l, b)$ has output $\tilde{\mathbf{o}}_m^{lb}$ on token $\tilde{\mathbf{x}}_m$, we define a global ablation intervention as the modification $\text{do}(\tilde{\mathbf{o}}_m^{lb} = \tilde{\mathbf{o}}_m^{lb} - \mathbf{h})$; and if head $(\ell, a)$ has input $\tilde{\mathbf{i}}_m^{\ell a}$ (after layer norm), a local ablation intervention is the modification $\text{do}(\tilde{\mathbf{i}}_m^{\ell a} = \tilde{\mathbf{i}}_m^{\ell a} - \mathbf{h})$. For boosting interventions, we implement $\text{do}(\tilde{\mathbf{o}}_m^{lb} = \tilde{\mathbf{o}}_m^{lb} + \mathbf{h})$ and $\text{do}(\tilde{\mathbf{i}}_m^{\ell a} = \tilde{\mathbf{i}}_m^{\ell a} + \mathbf{h})$.

Implementing the intervention depends on whether the token is a source or destination token in the downstream head. When the edge is a source edge ($t = $ source), we intervene in the source token ($\tilde{\mathbf{x}}_m = \tilde{\mathbf{x}}_j$), and we set $\mathbf{h}$ equal to the signal $\tilde{\mathbf{s}}^{\ell a, lb} = P_\mathcal{V} \mathbf{o}_m^{lb}$. In this case, $P_\mathcal{V}$ is the projector onto the $\mathcal{V}$ subspace defined by $S_{ij}^{\ell a}$. When the edge is a destination type edge, we intervene in the destination token ($\tilde{\mathbf{x}}_m = \tilde{\mathbf{x}}_i$), and we set $\mathbf{h}$ equal to the signal $\tilde{\mathbf{s}}^{\ell a, lb} = P_\mathcal{U} \mathbf{o}_m^{lb}$, where $P_\mathcal{U}$ is the projector onto the $\mathcal{U}$ subspace defined by $S_{ij}^{\ell a}$. In the case of local interventions, to make their encoding consistent with the assumptions imposed by layer norm folding, we center and scale them before adding them to the downstream input vector.

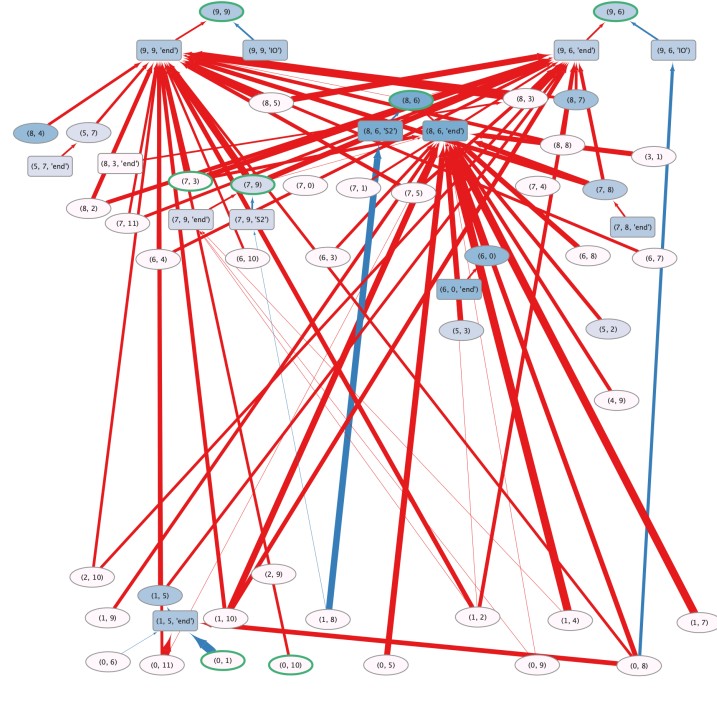

Figure 16: Skeleton of Trace Performed Using Residuals Directly Rather than Singular Vectors. Edges filtered to 170 occurrences or more.

**Magnitudes of Interventions.** Here we provide more detailed evidence of the magnitudes of the interventions we perform in §5.4. Given the intervened residual $\mathbf{x} + \mathbf{h}$ and the original residual $\mathbf{x}$, we use two metrics: the cosine similarity between $\mathbf{x}$ and $\mathbf{x} + \mathbf{h}$, and the ratio of the residual norms before and after the intervention $\frac{\|\mathbf{x}+\mathbf{h}\|}{\|\mathbf{x}\|}$.

For the single-edge interventions, we show the distribution across prompts that were intervened. In the case of the multi-edge interventions, different prompts can have different numbers of interventions. In that case, for each multi-edge intervention, we compute the metric per prompt, and the plots show the distribution of the average values across multiple multi-edge interventions. As expected, single edge interventions have smaller effects in the residual than multi-edge interventions. However, both are very peaked in the value 1.0, showing that most of these interventions have very small effects in the residual. See Figures 10, 11, 12, and 13 for more details.

**Effect of an Edge.** An edge in Figure 5 corresponds to a signal that has direct effect on a downstream attention head, and the weight of the edge corresponds to the strength of this signal. We expect that when we ablate an edge (for example), the attention score of the intervened token pair $(i, j)$ in the downstream head is decreased, especially for local interventions. In general this does not guarantee a particular indirect effect on model performance, due to indirect effects including self-repair and redundant paths in the circuit.

In Figures 17, 18, 19, and 20 we present examples showing the range of effects that edge ablation can have on both downstream attention scores, and on model performance. Edges in the figures are ordered by increasing weight (magnitude of the contribution (7) of the edge).

In all cases, edge ablations decrease attention scores as expected, with the effect varying depending on the weight of the edge. Further, in Figures 17, 19, and 20 edge ablations generally decrease model performance, again as expected. For the local ablations in Figure 20, the impact on model performance is proportionate to the decrease in attention score caused by the ablation. However in the case of global interventions, impact on model performance is not generally proportionate to the decrease in attention score.

Figure 18, corresponding to local ablations of edges into (9, 6) is a special case that we discuss in more detail below.

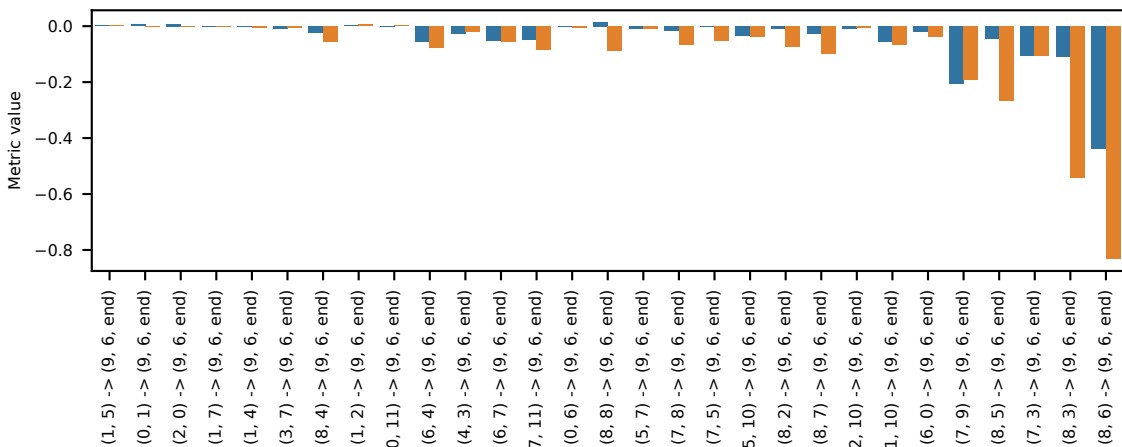

Figure 17: Global interventions effects on Edges into (9, 6, end). Blue: logit difference. Orange: attention scores difference.

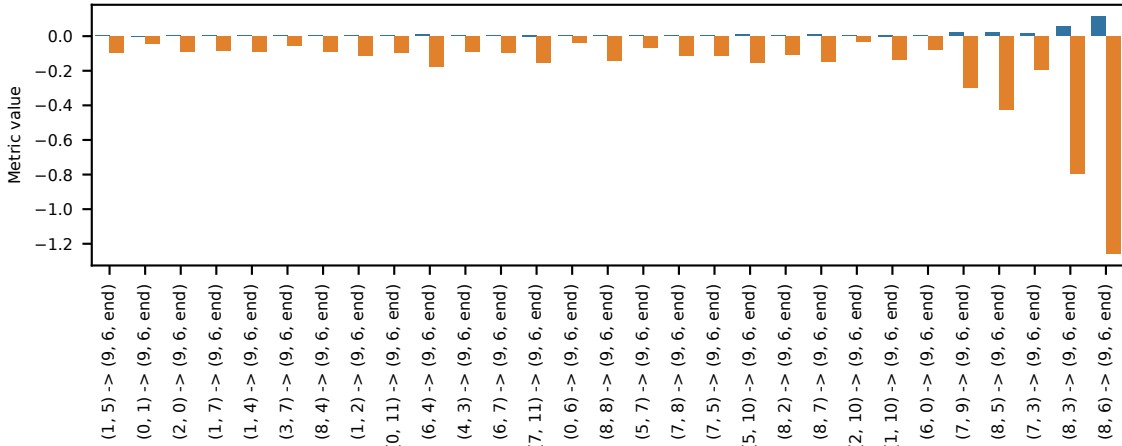

Figure 18: Local interventions effects on Edges into (9, 6, end). Blue: logit difference. Orange: attention scores difference. Note that while attention scores are decreasing, model performance is actually improving.

**Additional Intervention Results.** Here we show more extensive results of the intervention based validation experiments. First, we show that ablations generally decrease the IO logit, and boosting generally increases the IO logit. Figure 22, Figure 23, and Figure 24.

We also examine a wide range of multi-edge ablations to illustrate the impact of multiple paths in the traced circuit. First, note that Figure 14 gives a detailed look at the many redundant paths through the top layers of the traced circuit. In Figure 25 we show a variety of cases in which edges that are parts of parallel paths are ablated, both separately and together. In many cases we see evidence that the effect of ablating parallel paths has an additive nature. Then in Figure 26 we show a variety of cases in which edges that are parts of serial paths are ablated, both separately and together. In many of these cases, we see what is closer to a superposition effect, in which the effects of each edge are separately felt and not generally additive.

On a different point, in Figure 28 we show the effects of ablating edges outgoing from (2, 8). This shows that (2, 8) – which we identify as an important part of the IOI network in this paper, attending to the verb of the sentence – also has a causal effect on model performance.

**Ablation Results for (9, 6) Edges.** In §5.4 we show that ablating edges that feed into most components of the model decreases the value of $F$, measured as $F(X, e, \mathbf{x}) - F(X)$. Here we show that (9, 6) (a name mover head) exhibits the opposite behavior, but only for *local* interventions. Figure 21 shows the intervention results for the four most

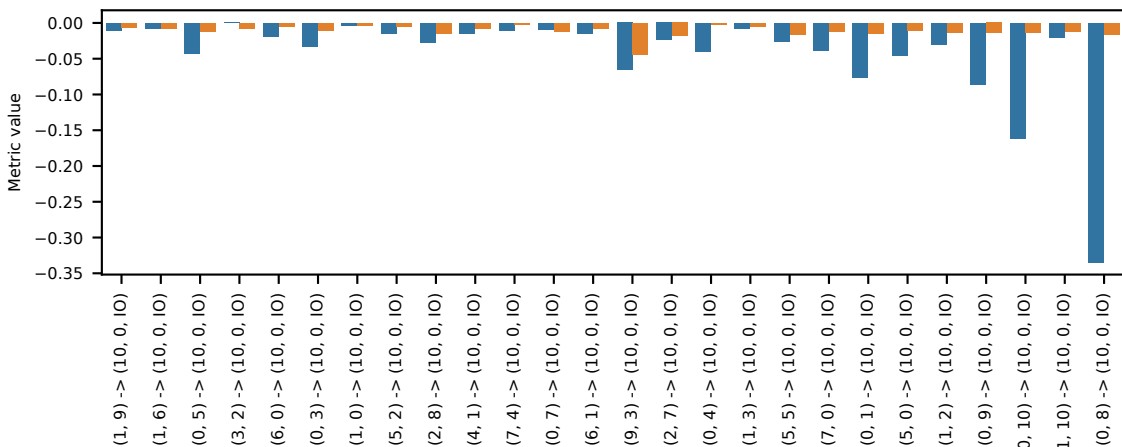

Figure 19: Global interventions effects on Edges into (10, 0, IO). Blue: logit difference. Orange: attention scores difference.

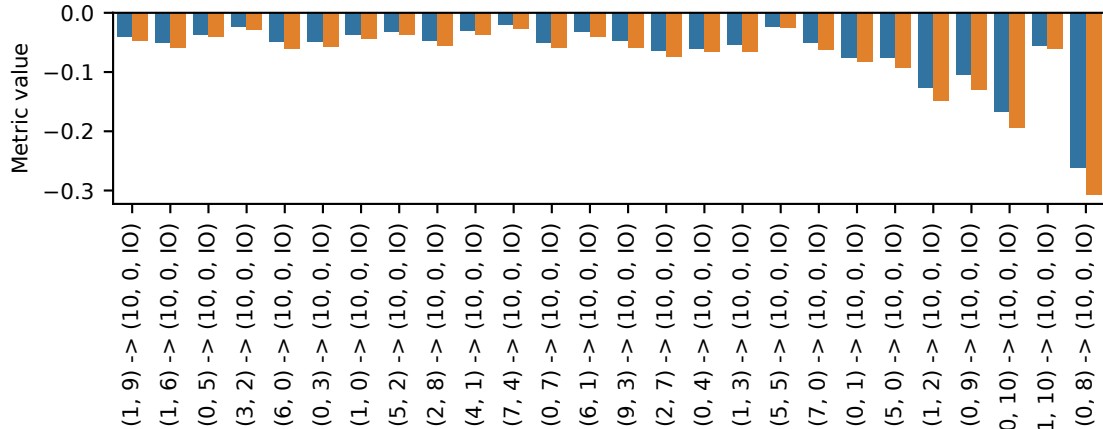

Figure 20: Local interventions effects on Edges into (10, 0, IO). Blue: logit difference. Orange: attention scores difference.

significant edges into (9, 6), and Figure 27 shows results for ablating all incoming edges of (9, 6). Note that global ablation of the (9, 6) signal decreases the logit of the IO token as expected. However, when the intervention is applied locally so that it only affects the (9, 6) head, ablation *increases* the logit of the IO token and boosting *decreases* the IO logit. More detail on ablations is provided in Figure 18. This figure shows that after ablation, the attention scores of the (9, 6) are indeed decreasing; it is only the downstream impact on the IO logit that is increasing. Further, the amount of increase of the IO logit is proportionate to the amount of decrease of the (9, 6) attention score. This suggests a different role for the (9, 6) compared to the other name mover heads. This effect is borne out across all local edge interventions in the (9, 6) and is suggestive of the need for further study.

**When Might the Sparse Decomposition Hypothesis *Not* Hold?**    Attention heads have been shown to have a variety of functions, not all of which correspond to testing low-dimensional subspaces. For example, some attention heads have the role of detecting when tokens $\mathbf{x}_i$ and $\mathbf{x}_j$ are identical (Elhage et al., 2021; Wang et al., 2023). In that case, we expect that $\mathbf{x}_i$ and $\mathbf{x}_j$ will have non-negligible inner products with most or all of the singular vectors of $\Omega$. In fact, we see exactly this phenomenon in the case of the duplicate-token head (3, 0) as shown in Figure 2(a).

**Proof of Lemma 1.**    Given vectors $\mathbf{x}$ and $\mathbf{y}$, among all rank-1 matrices having unit Frobenius norm, the matrix $D$ that maximizes $\mathbf{x}^\top D \mathbf{y}$ is $D = \frac{\mathbf{x}}{\|\mathbf{x}\|} \frac{\mathbf{y}^\top}{\|\mathbf{y}\|}$.

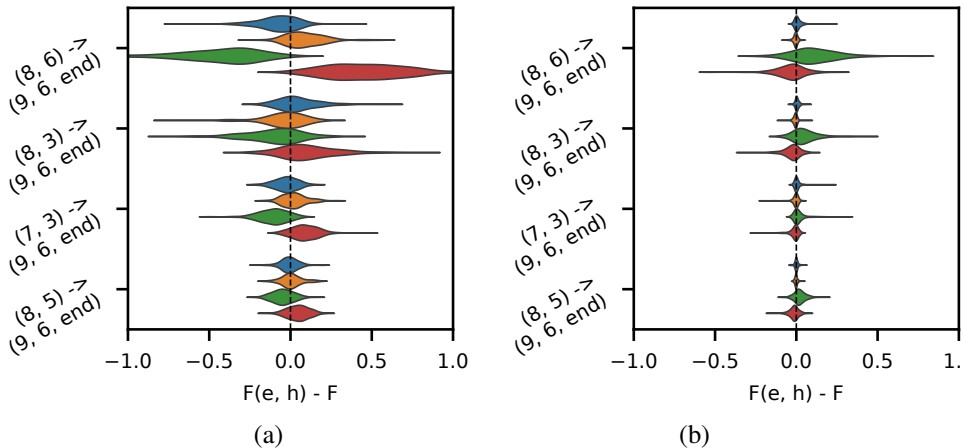

Figure 21: Intervention on Edges into (9, 6, end). (a) Global Intervention; (b) Local Intervention. Green: Ablation; Red: Boosting; Blue: Random Ablating; Orange: Random Boosting.

First we show that any rank-1 matrix having unit Frobenius norm can be expressed as the outer product of two unit-norm vectors. Consider a rank-1 matrix $X$ having unit Frobenius norm. Since $X$ is rank-1, we can write $X = \mathbf{x}\mathbf{y}^\top$. Now construct $\tilde{X} = \frac{\mathbf{x}}{\|\mathbf{x}\|}\frac{\mathbf{y}^\top}{\|\mathbf{y}\|}$. By construction $\tilde{X}$ is both rank-1 and unit norm. Matrices $X$ and $\tilde{X}$ differ by a constant factor $\frac{1}{\|\mathbf{x}\|\|\mathbf{y}\|}$. However, since they have the same norm, we must have $\|\mathbf{x}\|\|\mathbf{y}\| = 1$, and so $X$ can be expressed as the outer product of two unit vectors.

Next consider a unit-norm, rank-1 matrix $G = \mathbf{u}\mathbf{v}^\top$ for unit vectors $\mathbf{u}$ and $\mathbf{v}$. By way of contradiction, suppose $\mathbf{x}^\top G\mathbf{y} > \mathbf{x}^\top D\mathbf{y}$. Then $\mathbf{x}^\top \mathbf{u}\mathbf{v}^\top\mathbf{y} > \mathbf{x}^\top \frac{\mathbf{x}}{\|\mathbf{x}\|}\frac{\mathbf{y}^\top}{\|\mathbf{y}\|}\mathbf{y}$. The right hand side is the positive quantity $\|\mathbf{x}\|\|\mathbf{y}\|$. The left hand side is the product of the projections of $\mathbf{x}$ onto $\mathbf{u}$, and $\mathbf{y}$ onto $\mathbf{v}$. The product is maximized when $\mathbf{u} = \mathbf{x}/\|\mathbf{x}\|, \mathbf{v} = \mathbf{y}/\|\mathbf{y}\|$, or $\mathbf{u} = -\mathbf{x}/\|\mathbf{x}\|, \mathbf{v} = -\mathbf{y}/\|\mathbf{y}\|$. In either case, $\mathbf{x}^\top G\mathbf{y} = \mathbf{x}^\top D\mathbf{y}$, proving the lemma.

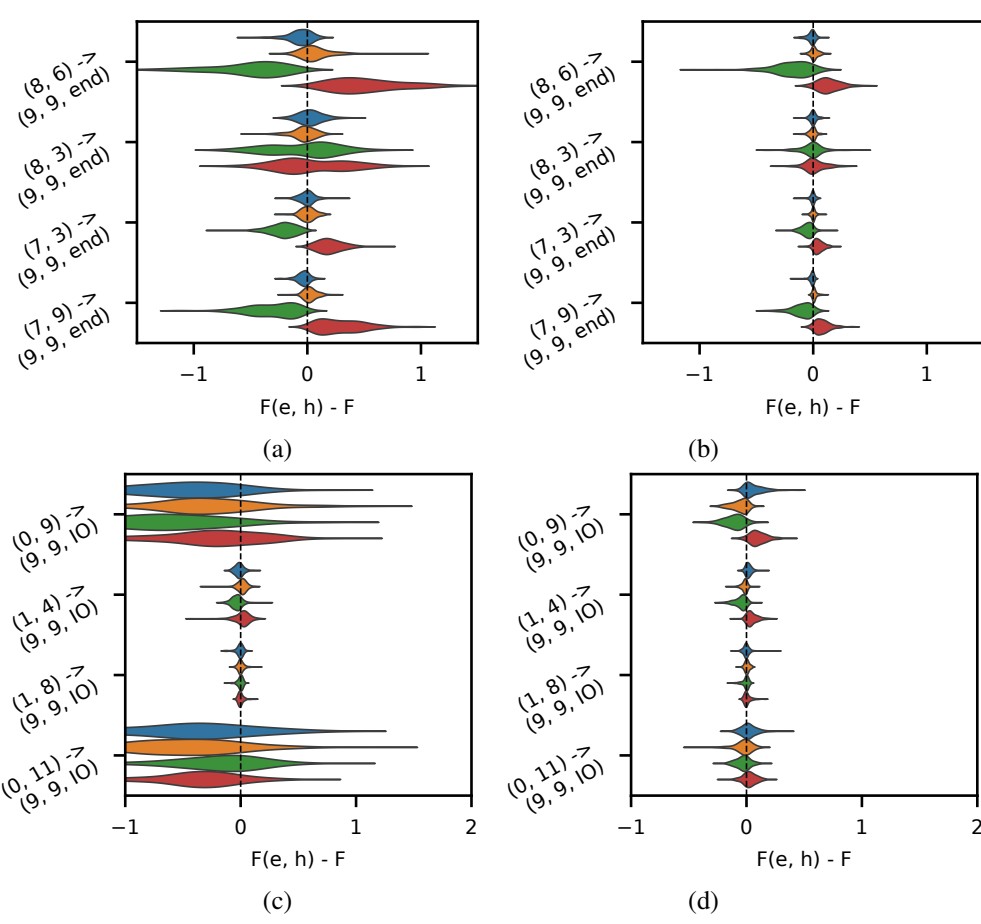

Figure 22: Edges into (9, 9): (a) (9, 9, end), Global intervention; (b) (9, 9, end), Local intervention; (c) (9, 9, IO), Global intervention; (d) (9, 9, IO), Local intervention. Green: Ablation; Red: Boosting; Blue: Random Ablating; Orange: Random Boosting.

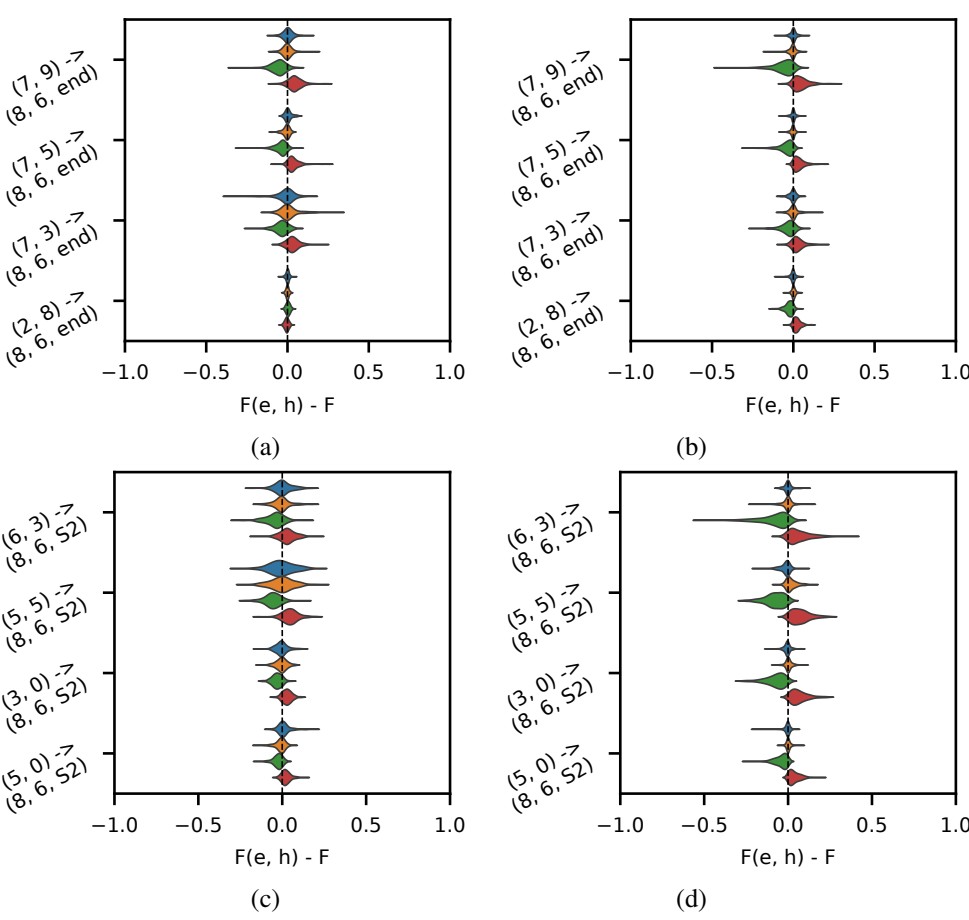

Figure 23: Edges into (8, 6): (a) (8, 6, end): Global intervention; (b) (8, 6, end): Local intervention; (c) (8, 6, S2): Global intervention; (d) (8, 6, S2): Local intervention. Green: Ablation; Red: Boosting; Blue: Random Ablating; Orange: Random Boosting.

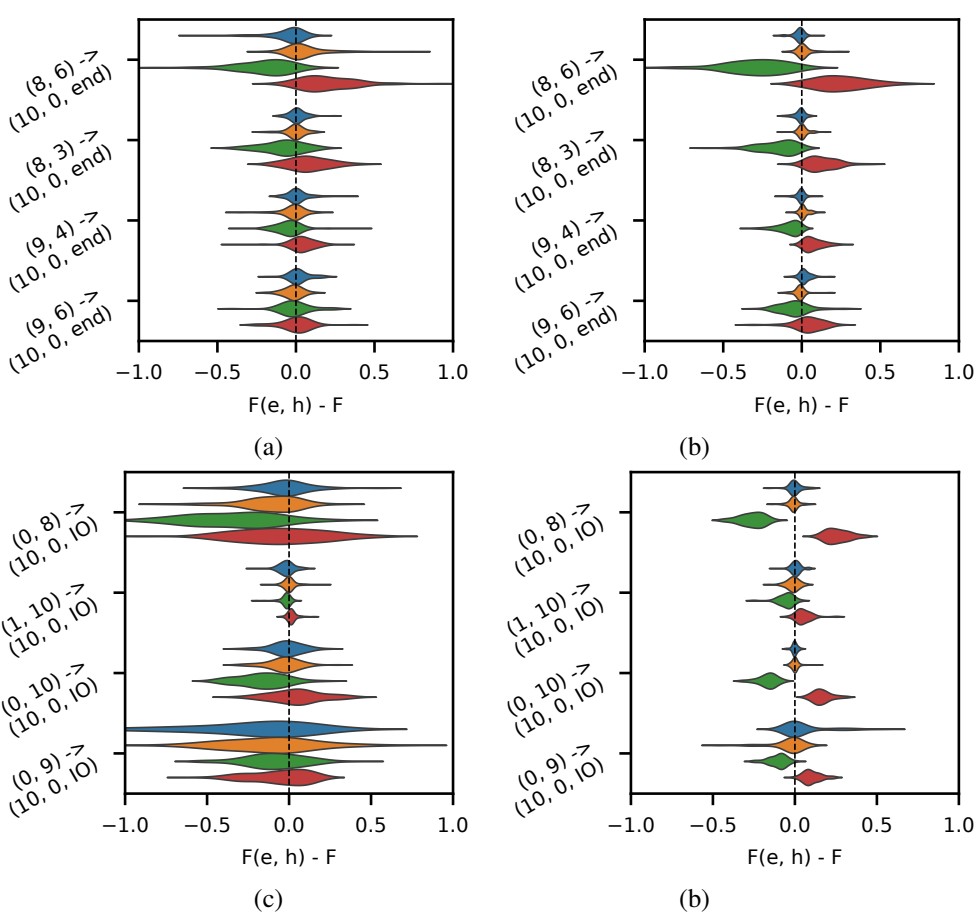

Figure 24: Edges into (10, 0): (a) (10, 0, end): Global intervention; (b) (10, 0, end): Local intervention; (c) (10, 0, IO): Global intervention; (d) (10, 0, IO): Local intervention. Green: Ablation; Red: Boosting; Blue: Random Ablating; Orange: Random Boosting.

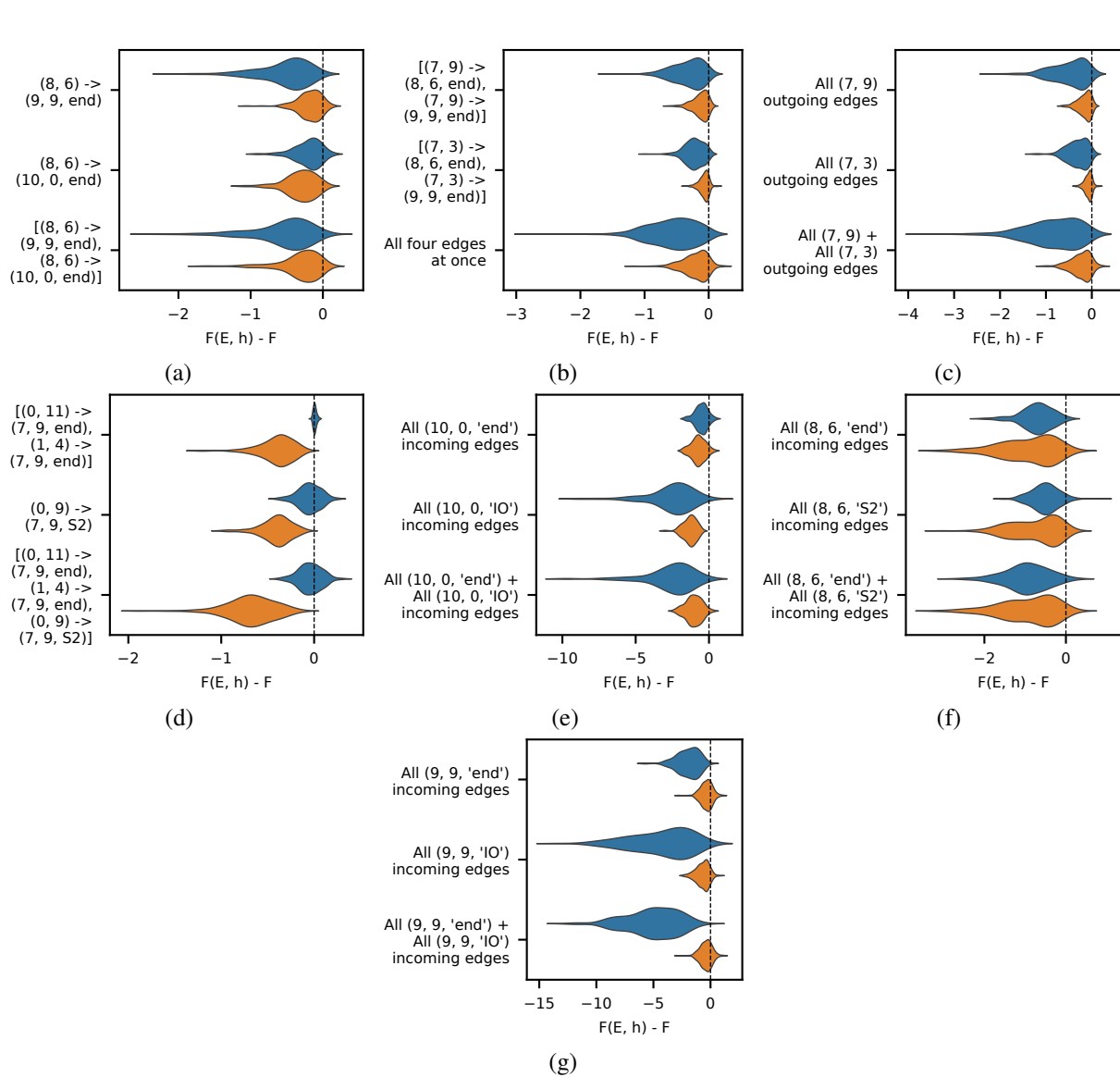

Figure 25: Parallel Multi-Edge Sets. Orange: Local Ablations; Blue: Global Ablations.

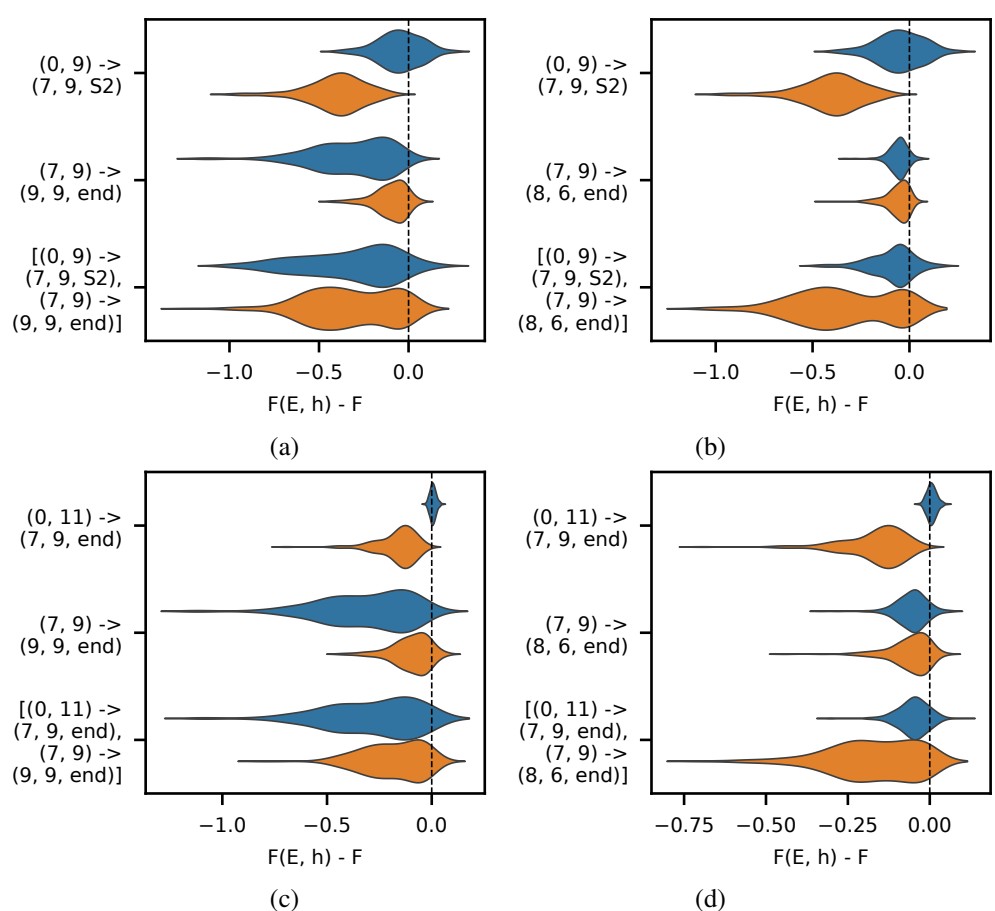

Figure 26: Serial Edge Sets. Orange: Local Ablations; Blue: Global Ablations.

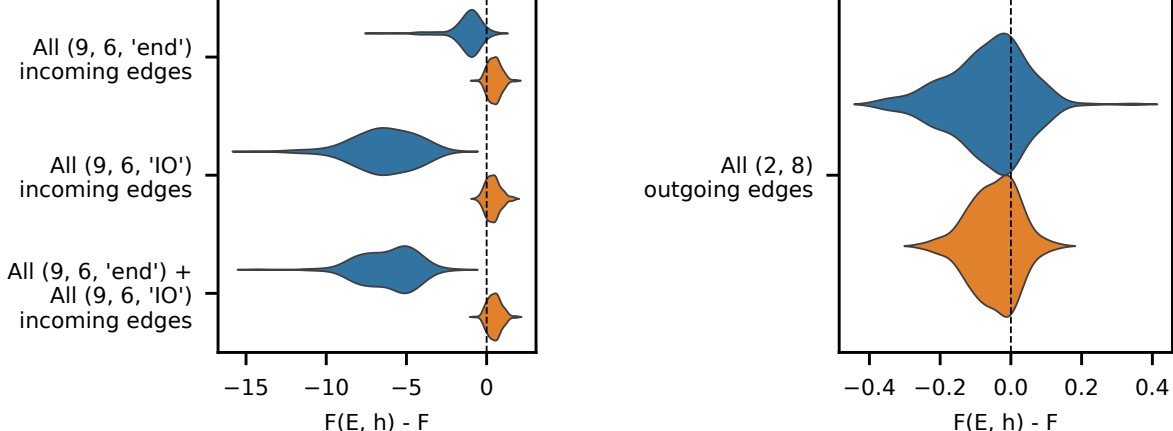

Figure 27: Multi Edge Sets for (9, 6).

Figure 28: Edges from (2, 8).

