# OpenReview forum: "Sparse Attention Decomposition Applied to Circuit Tracing"
_ICLR.cc/2025/Conference — Submitted to ICLR 2025_

### Official Review · Reviewer_CG35 · 2024-11-01

**Soundness:** 2
**Presentation:** 3
**Contribution:** 2
**Rating:** 3
**Confidence:** 4

**Summary:**

This work presents an approach for analyzing communication between attention heads in transformer models using SVD. It is shown that attention scores are often sparsely encoded in singular vectors of attention head matrices, enabling efficient isolation of features used for inter-head communication. The method is validated on GPT-2 small using the Indirect Object Identification (IOI) task, showing the redundant pathways and communication mechanisms between attention heads.

**Strengths:**

The main contribution of this paper lies in introducing a more scalable approach for interpreting the information flow within Transformers. Specifically,
- The use of SVD in circuit tracing seems simple and effective.
- The paper identifies new functionally important components (e.g., attention head 2,8) and provides a detailed analysis of redundant pathways in the model.
- Overall, the paper is well-structured and written.

**Weaknesses:**

The paper can be improved in several major aspects.
- The technical novelty seems limited. The idea of using dimensionality reduction (SVD in particular) to interpret and visualize models is not new.
- The study focuses on the attention layers. Do the MLP layers and layer normalization contribute to the change of causality relationships from layer to layer?
- The analysis is limited to a specific model GPT-2 small and a specific task (IOI). How do the findings generalize to other settings?
- Most of the findings are empirical. It's suggested to also explore why sparse decomposition occurs.
- Some of the study designs require further justification. For instance, the 70% threshold for filtering contributions seems arbitrary. Also, more justification is needed for the signal/noise separation approach.
- More discussion of failure cases where sparse decomposition might not hold would be valuable.
- The intervention focuses mainly on single-edge and simple multi-edge cases. What about more complex cases that involve multiple edges?
- There is limited comparison with other circuit analysis methods.

**Questions:**

See the detailed comments above.

---

> ### Author Response · Authors · 2024-11-19
> **Reviewer CG35 (1/2)**
>
> ### The technical novelty seems limited. The idea of using dimensionality reduction (SVD in particular) to interpret and visualize models is not new.
>
> It’s important to note that SVD is a general-purpose tool that can be used for many purposes.  Indeed, SVD has been used for certain tasks in analyzing LLMs in the past. However, we use **SVD to analyze models in an entirely new way**, one that has not been done before in analyzing LLMs to the best of our knowledge. As we say in the last paragraph of the introduction, and the last paragraph of the Related Work, what is important about our use of SVD is not that it decomposes attention matrices, but rather than it **provides a basis for decomposing attention inputs that expose their sparsity**.  As we write in the paper: **“We use SVD as a tool to decompose the computation of attention; the leverage we obtain comes from the resulting sparsity of the terms in the attention score computation.”**
>
> We also note that while some previous work has used SVD to interpret OV matrices and MLP weights, no previous work has used SVD in the analysis of QK matrices, as we do in the paper.
>
> ### The study focuses on the attention layers. Do the MLP layers and layer normalization contribute to the change of causality relationships from layer to layer?
>
> Regarding layer normalization, we do consider it. Quoting the paper (lines 243-245): **“We account for the effect of the layer norm using three techniques: weights and biases are folded into the downstream affine transformations, output matrices are zero centered, and the scaling applied to each token is factored into the contribution calculation.”**
>
> In fact we know from related work [1] that MLPs are not important for this task. However, as we discuss in the “Limitations” section, we plan to trace the importance of MLPs upstream for attention heads downstream by using exactly the same procedure described in the paper in Section 4.2 (essentially, checking if the MLP is “writing” in the directions that the attention head is “reading”).  This type of work is outside the current paper scope, but is on our roadmap for the near future.
>
> ### The analysis is limited to a specific model GPT-2 small and a specific task (IOI). How do the findings generalize to other settings?
>
> We are actively developing results for the Pythia model and for other tasks. Initial results are successful, and we plan to incorporate reference to them in the final paper.
>
> ### Most of the findings are empirical. It's suggested to also explore why sparse decomposition occurs.
>
> We agree that explaining why sparse decomposition occurs is important.  Indeed, we present an argument, based on known properties of how models encode concepts (eg, the linear representation hypothesis and the superposition phenomenon) for why sparse attention decomposition should be expected to occur.  This is the substance of Section 6 in our paper.
>
> ### Some of the study designs require further justification. For instance, the 70% threshold for filtering contributions seems arbitrary. Also, more justification is needed for the signal/noise separation approach.
>
> Indeed, when separating signal from noise, it is often the case that a threshold must be chosen.  Our choice of the 70% threshold is validated by the agreement we find in our results with the prior work of [1].   However, setting the threshold properly is worthy of further study.
>
> The signal/noise separation approach is a standard one in signal processing.   Almost all signal processing (eg, image/video compression, audio compression, etc) is based on finding a nearly-sparse encoding of the signal in an alternative orthogonal basis, and then zeroing out the small coefficients.  This allows for recovery of the “signal” without storing the “noise”.
>
> In our case, the alternative orthogonal basis is the set of singular vectors of the Omega matrix. This is why the demonstration of the sparsity of attention decomposition is so important: it allows a simple and effective (as we show) separation of signal from noise in the communication between attention heads.
>
> ### More discussion of failure cases where sparse decomposition might not hold would be valuable.
>
> Sparse attention decomposition will not hold in cases where the model has to use all the available orthogonal slices (in GPT-2 small there are 64) to reconstruct the attention score. However, we did not observe that in any of our experiments (eg Figure 3), and as we argue in Section 6, we generally do not expect this to happen.
>
> ### The intervention focuses mainly on single-edge and simple multi-edge cases. What about more complex cases that involve multiple edges?
>
> We provide examples of ablating multiple edges (as many as 10-12) at a time in our results.  We would be happy to perform ablation of more complex combinations of edges if the reviewer can suggest particular combinations that would expose useful validation for our model.

---

> ### Author Response · Authors · 2024-11-19
> **Reviewer CG35 (2/2)**
>
> ### There is limited comparison with other circuit analysis methods.
>
> We agree that this is a limitation, which we will add to the “limitations” section.  We note that direct comparison with other methods is difficult because our method finds circuits at a finer granularity than previous methods such as  (Wang et al 2023, Conmy et al 2023, Ferrando & Volta 2024).
>
> [1] Kevin Ro Wang, Alexandre Variengien, Arthur Conmy, Buck Shlegeris, and Jacob Steinhardt. Interpretability in the wild: a circuit for indirect object identification in GPT-2 small. In The Eleventh International Conference on Learning Representations, ICLR 2023, Kigali, Rwanda, May 1-5, 2023. OpenReview.net, 2023. URL https: //openreview.net/forum?id=NpsVSN6o4ul.

---

> > ### Comment · Reviewer_CG35 · 2024-11-29
> >
> > Thanks for the authors' response, which clarifies some of my questions and acknowledges my concerns (e.g., technical novelty, parameter setting, comparison with baselines). I'll thus keep my score.

---

### Official Review · Reviewer_ibeQ · 2024-11-03

**Soundness:** 3
**Presentation:** 3
**Contribution:** 3
**Rating:** 6
**Confidence:** 3

**Summary:**

This paper introduces a method based on sparse attention decomposition for analyzing the communication and coordination among attention heads in Transformer models. By constructing attention scores sparsely in a new basis through Singular Value Decomposition (SVD), we identify key communication paths between attention heads within the model. Experiments demonstrate that the communication paths identified through sparse decomposition have a practical causal effect on model functionality, enhancing the model's interpretability and offering new insights for understanding and improving Transformer models.

**Strengths:**

1, Novelty: This paper addresses the previously challenging issue of identifying and interpreting the complex interactions between attention heads in Transformer models by proposing a novel SVD-based sparse decomposition method.

2, Interpretability: The paper uncovers the communication pathways between attention heads in Transformer models, enhancing researchers' understanding of the model's internal workings.

**Weaknesses:**

1, Possible computational complexity: The computation of SVD and sparse decomposition is usually very complex and requires a lot of computing resources. What is the computational complexity and computing resources consumed in this paper? Have the factors related to computational complexity and required computing resources been considered?

2, Lack of open source code: The author should provide source code to facilitate others to reproduce and verify.

**Questions:**

The computation of SVD and sparse decomposition is usually very complex and requires a lot of computing resources. What is the computational complexity and computing resources consumed in this paper? Have the factors related to computational complexity and required computing resources been considered?

---

> ### Author Response · Authors · 2024-11-19
> **Reviewer ibeQ**
>
> ### Lack of open source code: The author should provide source code to facilitate others to reproduce and verify.
>
> Thank you for your suggestion. We did not provide a link to our code due to anonymity requirements. Instead, we are providing a .zip file containing the code to reproduce all the results (see Supplementary Material). Please check it out and let us know if you have any questions or suggestions about the code.
>
> ### The computation of SVD and sparse decomposition is usually very complex and requires a lot of computing resources. What is the computational complexity and computing resources consumed in this paper? Have the factors related to computational complexity and required computing resources been considered?
>
> Thank you for your comments. In fact, the computational complexity of the SVDs we use is not very great;  we are able to run all the SVDs needed for our experiments in less than a minute on a Mac M1 Max laptop.   Because of the low computational cost, we did not mention it in the paper.
>
> We need only one SVD per attention head in the model, and there are 144 heads in the model.  For the case of GPT 2, the Omega matrix is 769 x 769, with rank 64 (which makes the cost even cheaper). Our method also required only one forward pass in the model. Compared with other approaches, such as path patching (used by [1]), our method is much more efficient.

---

### Official Review · Reviewer_V5Fk · 2024-11-04

**Soundness:** 3
**Presentation:** 2
**Contribution:** 2
**Rating:** 5
**Confidence:** 4

**Summary:**

The authors explored some of the technical details of GPT-2 through SPARSE ATTENTION DECOMPOSITION. Their tracing study reveals considerable detail not present in previous studies, shedding light on the nature of redundant paths present in GPT-2. Their traces go beyond previous work by identifying features used to communicate between attention heads when performing IOI.

**Strengths:**

(1) The theoretical proofs in this paper are remarkable.
(2) The figures and tables in the paper are visually appealing, which enhances readability to a certain paper.
(3) The related work and literature survey are adequate and well organized.

**Weaknesses:**

(1) My big concern is that the paper may be technically obsolete. More mainstream experiments are now being conducted in GPT 4o and GPT o1-based settings. I don't understand why the authors are still conducting experiments on GPT 2. The gap between GPT 2 and GPT 4o and GPT o1-based methods is huge, so I think the experiments and the motivation are very limited, and the techniques in the paper may not be valid for the GPT 4 and GPT o1-based settings.
(2) The writing of the article is obscure. Maybe this article is hard to understand and follow. Reading through the entire paper, I'm not sure what the focus of the article FOCUSED on.
(3) The topics “SPARSE ATTENTION DECOMPOSITION” and “ Circuit Tracing ” did not attract widespread interest, and the importance of this area was not emphasized.
(4) In sum, our contributions are twofold. First, we draw attention to the fact that attention scores are typically sparsely decomposable given the right basis. This has significant implications for the interpretability of model activations. Why? The authors' experiments did not prove their interpretability.
(5) The paper was not compared to multiple state-of-the-art BASELINE methods, so there is insufficient validation of its effectiveness. For example, no quantitative comparison results can be seen in Figures 1, 2, and 3.
(6)The interpretability of the paper is assessed by the "Contribution" to attn. score"?

**Questions:**

See the  Weaknesses.

**Details Of Ethics Concerns:**

No Comments.

---

> ### Author Response · Authors · 2024-11-19
> **Reviewer V5Fk (1/2)**
>
> ### (1) My big concern is that the paper may be technically obsolete. More mainstream experiments are now being conducted in GPT 4o and GPT o1-based settings. I don't understand why the authors are still conducting experiments on GPT 2. The gap between GPT 2 and GPT 4o and GPT o1-based methods is huge, so I think the experiments and the motivation are very limited, and the techniques in the paper may not be valid for the GPT 4 and GPT o1-based settings.
>
> Thanks for your comments.  It’s important to understand that our work applies to open-source models. Indeed, the entire mechanistic interpretability community uses only open-source models, because it is necessary to have access to model internals (weights) in order to uncover internal mechanisms.   Hence, models such as GPT 4o are not suitable for mechanistic interpretability studies such as ours. You can see in the references that we cite (eg, Conmy 2023, Ferrando 2024, Geiger 2024, Gurnee 2024, Harra 2023, …) that they all use open-source models.
>
> Among open-source models, GPT-2 is particularly well-suited for studies such as ours because it exhibits interesting behavior (eg, good performance on the IOI task we study) while being small enough to allow for deep understanding. That is why the studies we compare to (Wang et al 2023, Conmy et al 2023, Ferrando & Volta 2024) all also use GPT-2 for their studies.
>
> ### (2) The writing of the article is obscure. Maybe this article is hard to understand and follow. Reading through the entire paper, I'm not sure what the focus of the article FOCUSED on.
>
> We are eager to improve the clarity of focus in the paper. To make clear, we list our contributions as the final paragraph of the introduction: first, we expose an important property of transformer-based models that has not previously been appreciated:  attention scores (the heart of the transformer mechanism) are actually sparsely encoded.  This means that one can identify what signals are passing between attention heads when they fire – opening up a large source of insight into how these models work.  Second, we show the power of sparse decomposition by using it to trace a “famous” circuit in GPT-2, in a manner that is much faster and more thorough that has been in any previous work.
>
> ### (3) The topics “SPARSE ATTENTION DECOMPOSITION” and “ Circuit Tracing ” did not attract widespread interest, and the importance of this area was not emphasized.
>
> Indeed, Sparse Attention Decomposition is a new phenomenon, one that opens up important sources of insight in analyzing transformer-based models;  as such, the term does not appear in the literature to date (to the best of our knowledge). “Circuit tracing” is a kind of study that is of great interest in the mechanistic interpretability literature: (Wang et al 2023, Conmy et al 2023, Ferrando & Volta 2024) are all circuit-tracing papers.
>
> ### (4) In sum, our contributions are twofold. First, we draw attention to the fact that attention scores are typically sparsely decomposable given the right basis. This has significant implications for the interpretability of model activations. Why? The authors' experiments did not prove their interpretability.
>
> Thank you for pointing out that this sentence could be supported more clearly.   We describe the implications for interpretability in our response to point (6) below and will add these remarks to the paper for clarity.
>
> ### (5) The paper was not compared to multiple state-of-the-art BASELINE methods, so there is insufficient validation of its effectiveness. For example, no quantitative comparison results can be seen in Figures 1, 2, and 3.
>
> Regarding the comparison, we made a direct comparison of our circuit with the circuit found in [1].  This prior work [1] is also used as comparison in other studies, eg,  (Conmy et al 2023, Ferrando & Volta 2024).  However, we note that Figures 1, 2, and 3 do not relate to the circuit we trace, but rather serve to document and explain the phenomenon of sparse attention decomposition, which is a fundamental contribution of our paper that is separate from the circuit tracing result.

---

> ### Author Response · Authors · 2024-11-19
> **Reviewer V5Fk (2/2)**
>
> ### (6)The interpretability of the paper is assessed by the "Contribution" to attn. Score"?
>
> Contribution to attention score is a new, causal relationship that we define and develop in our paper.  Contribution is a direct measure of how an upstream attention head causes a downstream head to fire, and what signal is sent to cause that head to fire.  Hence, the contribution provides interpretation of the mechanism at work inside the transformer model.
>
> We show interpretability of signals at the end of Section 5.2 and in the Appendix.  More generally, we show the interpretability of contributions through the causal intervention validations in Section 5.  In that section we show that model performance can be directly improved, or directly impaired, by increasing or decreasing the contributions we uncover.
>
> [1] Kevin Ro Wang, Alexandre Variengien, Arthur Conmy, Buck Shlegeris, and Jacob Steinhardt. Interpretability in the wild: a circuit for indirect object identification in GPT-2 small. In The Eleventh International Conference on Learning Representations, ICLR 2023, Kigali, Rwanda, May 1-5, 2023. OpenReview.net, 2023. URL https: //openreview.net/forum?id=NpsVSN6o4ul.

---

### Meta-Review · Area_Chair_9ZxC · 2024-12-05

**Metareview:**

The paper presents an approach for analyzing communication between attention heads in transformer models using SVD, but reviewers raised several critical concerns. Reviewers noted the limitation of the analysis, which is confined to GPT-2 and does not consider other mainstream models. Additionally, the paper lacks a comparison with other circuit analysis methods. Given these issues, I recommend rejection.

**Additional Comments On Reviewer Discussion:**

To response to Reviewer V5Fk, the authors stressed that their work applies to open-source models, so models such as GPT 4o are not suitable. However, the work does not  conducting experiments on other open-source models such as Llama. Besides, the authors acknowledges their limited comparison with other baseline methods.

---

### Decision · Program_Chairs · 2025-01-22

Reject